



# Data-driven probabilistic surrogate model for floating wind turbine lifetime damage equivalent load prediction

Deepali Singh[1], Erik Haugen[2], Kasper Laugesen[2], Richard P. Dwight[1], and Axelle Viré[1]

[1]Delft University of Technology, Kluyverweg 1, 2629 HS Delft, The Netherlands
[2]Siemens Gamesa Renewable Energy, Tonsbakken 16, 2740 Skovlunde, Denmark

**Correspondence:** Deepali Singh (d.singh-1@tudelft.nl)

**Abstract.**

Floating offshore wind turbines experience complex hydrodynamic and aerodynamic loading influenced by substructure types and stochastic environmental conditions. Accurately estimating the lifetime fatigue loads requires analyzing thousands of operational scenarios, leading to high computational costs. Moreover, choosing the right input features driving fatigue in floating wind systems and appropriately binning them still remains an open question. We present a fast probabilistic surrogate that maps the site conditions to the loads on the wind turbine. The probabilistic aspect allows the propagation and quantification of statistical uncertainties from the stochastic input quantities on the resulting loads. A fast surrogate eliminates the need to fit a distribution to the site conditions or bin the input data. Rather, all available met-ocean data can be directly used as input, which automatically accounts for the joint distribution in the calculations. The surrogate model in this study uses the mixture density network (MDN) to predict the conditional distribution of the 10-minute damage equivalent loads (DELs) for a 6 MW spar-type floating wind turbine. The MDN achieves high accuracy ($R^2 > 0.99$) in capturing DEL means while efficiently propagating the statistical uncertainties. Furthermore, the surrogate enables quick estimation of 25-year lifetime fatigue damage across a range of potential floating wind farm sites, demonstrating its capability to facilitate rapid decision-making during preliminary site analysis.

## 1 Introduction

Floating offshore wind turbine (FOWT) technology has witnessed a surge in research interest in recent years following the rapidly increasing demands for renewable power production. The structural response of a FOWT is a crucial indicator of its performance, safety, and reliability. During its operational lifetime, a FOWT accumulates fatigue damage as it undergoes time-variable loading in response to the complex and stochastic marine environment. The nature, magnitude, and extent of fatigue are unique to the type of floating foundation, mooring line configuration, wind turbine material, control algorithms, and site conditions.

To ensure a safe and reliable operational life, the FOWT undergoes a certification process involving a rigorous analysis of various design load cases (DLCs) defined by the International Electrotechnical Commission in IEC 61400-3-2 (IEC, 2024a). The first step involves simulating the DLCs on a type-certified rotor-nacelle assembly with a reference tower and floating foun-



dation. More detailed information about the site is included while defining the DLCs as the project progresses. Subsequently, a site-specific tower, foundation, and mooring line configuration are defined and a site-specific certification study is performed. The calculations are typically made using time-domain multi-physics engineering tools (Jonkman, 2013; Larsen and Hansen, 2007; Couturier and Skjoldan, 2018; Skjoldan, Peter Fisker, 2011) throughout this process.

    Fatigue is a multi-scale phenomenon that depends on the material composition, composite structure, geometry, and inflow

dynamics. The estimation of the fatigue damage for FOWTs, in particular, is computationally intense. The lifetime fatigue load assessment entails calculating the 10-minute damage equivalent loads (DELs) on multi-variate bins of typical variables characterizing the site and scaling them to the observed probability of occurrence. Not all site variables can be practically included in fatigue load analysis, as the required number of simulations increases exponentially with each additional variable. The choice of the variables in the offshore environment that have the most impact on FOWT fatigue is currently an active

area of research (Papi and Bianchini, 2024). The total computational cost of the simulations also constrains the lower limit of the bin size. While industry-standard engineering tools are necessary for certification, the preliminary site analysis can benefit from *data-driven surrogate* models to provide quick load estimates. Data-driven surrogates can infer complex relationships from data observations alone and do not require prior knowledge of the underlying physics. Fatigue, which is difficult to model using lower-fidelity physics-based approaches, can benefit from such data-driven methods. Using surrogates that can accurately

predict DELs can potentially eliminate the need to bin the site data, fit a multi-variate joint distribution to it, or limit the total number of parameters. Once trained, surrogate models can directly use all the available site information to estimate the site-specific DELs quickly. In addition, probabilistic surrogates can also propagate the statistical uncertainty from the stochastic input variables to the loads.

    Data-driven surrogates for wind turbine or wind farm level loads are often designed with deterministic models. Given a

training dataset with $d$-dimensional input parameters $\boldsymbol{x} \in \mathbb{R}^d$, the deterministic surrogate maps them to the corresponding output observations $y \in \mathbb{R}$. However, the assumption of a deterministic relationship between inputs and outputs does not hold in our case. For instance, keeping every other input constant, a single value of 10-minute mean wind speed can correspond to an infinite number of turbulent inflow patterns, resulting in an infinite number of DEL values with a certain probability distribution conditioned on that wind speed. Probabilistic surrogates model the statistical uncertainty in the input variables by representing

them as a random variable $\boldsymbol{X}$ with a joint probability density function (pdf). The corresponding output is, therefore, also a random variable denoted as $Y$. The standard Gaussian process regression (GPR) (Rasmussen and Williams, 2006) is one such probabilistic surrogate that is capable of uncertainty quantification. However, in its standard form, it is restricted to normally distributed homoscedastic responses. Nevertheless, due to its flexibility and ease of implementation, it is widely used as a surrogate to estimate the fatigue load response in wind turbines (Teixeira et al., 2017; Avendaño-Valencia et al., 2021; Li and

Zhang, 2019, 2020; Gasparis et al., 2020; Dimitrov et al., 2018; Slot et al., 2020).

    Further interest in quantifying the uncertainty of the short-term fatigue loads as a function of the input parameters has initiated research into heteroscedastic surrogates. *Heteroscedasticity* refers to the heterogeneity in the response variance as a function of the inputs. The variance observed in DEL at the tower bottom at, for instance, very large values of significant wave height is generally larger than that in calm ocean conditions. It is, therefore, an important consideration when choosing



the appropriate surrogate modeling approach for load uncertainty quantification. Murcia et al. (Murcia et al., 2018) use 100 turbulent inflow realizations at each sample point to obtain the first two moments of the fatigue response. Thereafter, they create two independent surrogates using Polynomial Chaos Expansion (PCE) to model the mean and standard deviation of the fatigue loads on the DTU 10MW reference wind turbine. Even though they use only 140 training samples for their model, the replications scale the computational cost by 100, eventually leading to a costly training database. Another replication-based approach is taken by Zhu et al. (Zhu and Sudret, 2020) to model the load response using generalized lambda distributions. In this study, 50 inflow wind field realizations are used at each input sample to estimate the four lambda parameters. Four PCE surrogates are then used to model the parameters independently. The main drawback of replication-based methods is the cost of generating the training database, which makes it challenging to apply them to computationally demanding applications such as floating wind turbines. Secondly, the goodness of fit relies heavily on estimating the statistical parameters in the first step.

Heteroscedasticity can also be modeled using statistical methods. Zhu et al. (Zhu and Sudret, 2021) extend the replication-based approach to derive a statistical method combining generalized least-squares with maximum conditional likelihood to estimate the lambda parameters without replications. The main advantage of this method is that it does not assume a Gaussian distribution. However, it cannot handle multi-modality. Abdallah et al. (Abdallah et al., 2019) use parametric hierarchical Kriging to predict blade-root-bending-moment extreme loads that are heteroscedastic on a 2MW onshore wind turbine. Their approach combines low- and high-fidelity observations, where the low-fidelity model informs the high-fidelity GPR. They show that introducing hierarchy helps make the model selection process more robust than the manual tuning of GPR parameters. Singh et. al (Singh et al., 2022) apply chained GPR that uses variational inference within a Bayesian framework to account for heteroscedasticity in the data and make predictions of site-specific load statistics on a more complex case of offshore wind turbines. The model can capture the heteroscedasticity in a small dataset but is not scalable to high dimensional problems. To address the scalability constraints, the authors extend the study to use mixture density networks on the same dataset to quantify the uncertainty in the load response (Singh et al., 2024a). The application of probabilistic surrogates to floating wind turbines has only been studied to a limited extent in literature. Li et al. (Li and Zhang, 2019) model the uncertainty in the site conditions using a C-vine copula combined with ANN and GPR.

In summary, only a few approaches attempt to model the uncertainty in the load response of the turbine and the tower. Of those that do, only some consider complex offshore floating systems. Following the promising performance of mixture density networks (MDNs) for fixed bottom wind turbines (Singh et al., 2024a), in this study, we aim to extend the framework to a more complex application of a spar-buoy type wind turbine case. In the case of MDN, the target is modeled as a mixture of $m \in \mathbb{N}$ Gaussian kernels of varying proportions, capable of generating complex distributions when combined. MDN uses feed-forward networks to learn the parameters of the mixture model.

The main objective of this study is to present a probabilistic data-driven surrogate modeling framework that maps 10-minute statistics of the environmental conditions to the corresponding conditional probability distribution of the DEL on the floating spar buoy with a 6MW Siemens Gamesa wind turbine. The DEL values are calculated using the Siemens Gamesa in-house tool that couples the aeroelastic code, BHawC (Skjoldan, Peter Fisker, 2011), with the hydrodynamic solver, OrcaFlex (Arramounet et al., 2019). A schematic of this mapping is shown in Figure 1. A highly flexible probabilistic machine learning approach for





the surrogate, the mixture density network (MDN) (Bishop, 1994) is used in this study. The probabilistic estimates of DELs
are used to subsequently calculate the lifetime fatigue loads at various potential floating wind sites.

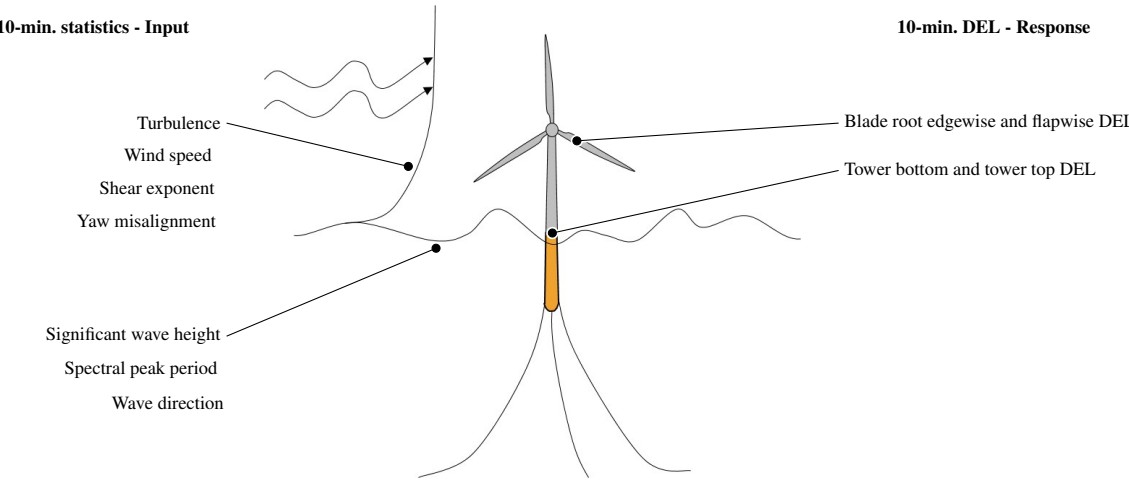

**Figure 1.** Schematic of the surrogate modeling objective.

The rest of the paper is structured as follows. In Section 2, we introduce the floating wind turbine model, describe the
simulation tools used, and outline the input features, including their ranges and the sampling strategy. Section 3 then presents
the theoretical foundation of the Mixture Density Network (MDN), discusses the chosen hyperparameters, and explains the
criteria for evaluating model performance. The results, presented in Section 4, are divided into three sub sections. First, we
analyze how the model's performance converges across a range of training samples. Next, we validate the selected MDN
model's 10-minute conditional distribution estimates under randomly chosen operating conditions, comparing them with those
generated by BHawC. Finally, we demonstrate how the probabilistic 10-minute estimates can be used to propagate the statistical
uncertainty to the lifetime fatigue damage. Concluding remarks are provided in Section 5.

## 2 Setup

### 2.1 Definition of the floating wind turbine

The floating wind turbine in this study is based on a modified geometry of the Hywind Scotland spar buoy foundation (Equinor
ASA, 2022). It comprises a 6MW Siemens Gamesa Renewable Energy direct drive wind turbine assembly, SWT-6.0-154,
mounted on a spar buoy. The characteristic wind turbine parameters are listed in Table 1. The simulations use a tower with a
larger diameter than the tower designed for the Hywind Scotland site. It is, therefore, stiffer and has a higher natural frequency
than its installed counterpart. The geometry details of the tower and the floating platform used in the simulation are provided
in Table 2. The floating substructure is attached to the ocean floor using catenary mooring lines, equally spaced at $120°$ in





**Table 1.** Parameters of 6MW Siemens Gamesa wind turbine.

| Parameter | Property |
| --- | --- |
| Rated power | 6000kW |
| Configuration | 3-bladed |
| Power control | Pitch |
| Drivetrain | Direct drive |
| Rotor diameter | 154m |
| Hub height | 96m |
| Rated wind speed | $12\text{ms}^{-1}$ |
| Rated tip speed | $89\text{ms}^{-1}$ |
| Nacelle mass | 360 te |

crowfoot configuration using bridle lines as shown in Figure 2 (Equinor ASA, 2022). The structural properties of the main mooring lines and the bridle lines are listed in Table 3.

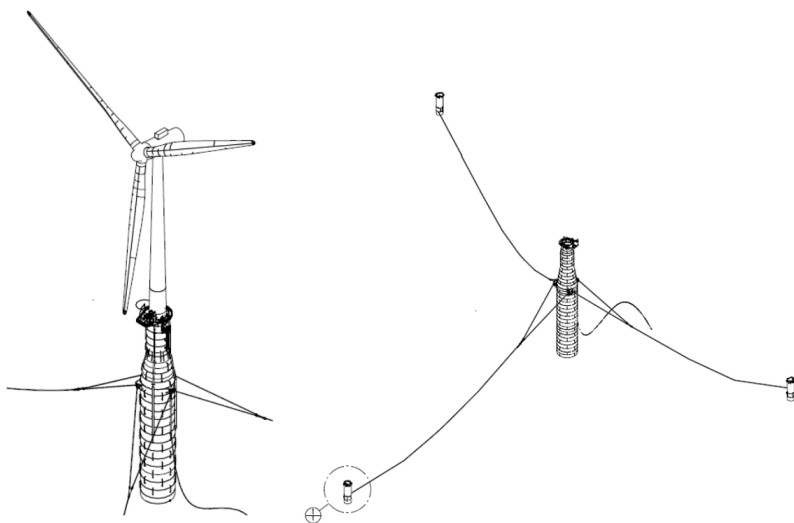

**Figure 2.** Hywind Scotland spar buoy with crowfoot mooring line configuration (Equinor ASA, 2022).

**2.2 Numerical model**

The damage equivalent loads are obtained through time-domain hydro-servo-aeroelastic simulations performed using BHawC-OrcaFlex coupled implementation. BHawC has been used for several years at Siemens Gamesa for wind turbine load calculations and is continuously validated against turbine prototypes and the entire operational fleet. Similar analysis may be performed with OpenFAST (NREL, 2022; Jonkman, 2013) coupled with OrcaFlex (Masciola et al., 2011) via FASTLink for



**Table 2.** Wind turbine tower and foundation properties. (Busse-makers, 2020; Equinor ASA, 2022; Equinor, 2024)

| Parameter | Value |
| --- | --- |
| Tower bottom outer diameter | 9.45m |
| Tower bottom thickness | 0.08m |
| Tower top outer diameter | 4.89m |
| Tower top thickness | 0.029m |
| Tower bottom elevation above SWL | 13m |
| Draft | 78m |
| Platform length Platform top geometry- length | 12m |
| Platform top geometry - diameter | 9.4m |
| Platform taper length | 15m |
| Platform bottom geometry - length | 58m |
| Platform bottom geometry - diameter | 14.4m |

**Table 3.** Catenary mooring line properties.

| Parameter | Value |
| --- | --- |
| Number of mooring lines | 3 |
| Angle between mooring lines | 120° |
| Mooring bridle line length | 50m |
| Mooring bridle line mass per unit length | 0.348te/m |
| Mooring main line length | 610m |
| Mooring main line mass per unit length | 0.4322te/m |
| Mooring line anchor radius | 640m |

reproducibility. Arramount et al. (Arramounet et al., 2019) present the mathematical background for the software coupling. In short, the tower, the rotor nacelle assembly, and the blade elements are dynamically modeled in BHawC. The BHawCLink module acts as a communication channel with the dynamic link library, connecting it to the OrcaFlex API. OrcaFlex simulates the hydrodynamic response of the floater element. Time integration is performed individually on both elements while accounting for the response of the other structure per iteration (Arramounet et al., 2019).

The inflow turbulence is modeled using a spatially varying frozen wind field based on the Mann model (Mann, 1998). The tangential and axial induced velocities are calculated on several aerodynamic nodes on the blades using the blade element momentum theory coupled with Prandtl's tip loss correction and thrust correction at high induction values. Skewed and unsteady inflow is modeled using the method introduced by Björck et al. (Björck, 2000).

The structural elements are modeled using the co-rotational formulation providing geometric nonlinearity (Rubak and Petersen, 2005). The tower, shaft, and blade substructures are modeled using beam elements. The Torsethaugen two-peak wave spectrum generates swell and local wind-driven waves (Torsethaugen and Haver, 2004). The various elements of the OrcaFlex model are shown in Figure 3. A 6-DOF rigid buoy in OrcaFlex represents the floating substructure. The mooring lines are modeled in OrcaFlex. Each line is divided into several massless spring segments, joined by elements with lumped properties such as mass, damping, added mass, buoyancy, and material properties.

The simulations are initialized in BHawC with a quasi-static approach where the environmental loads (wind and waves) and inertial loads (gravity and buoyancy) are slowly ramped up in small steps. For every load step, OrcaFlex determines the mooring line static equilibrium based on the floater position determined by BHawC. BHawC calculates the global equilibrium position based on the stiffness matrices and interface loads provided by OrcaFlex. Once the global equilibrium is calculated, the next load step is applied. The dynamic part of the simulations consists of an initialization phase of 300s to eliminate any




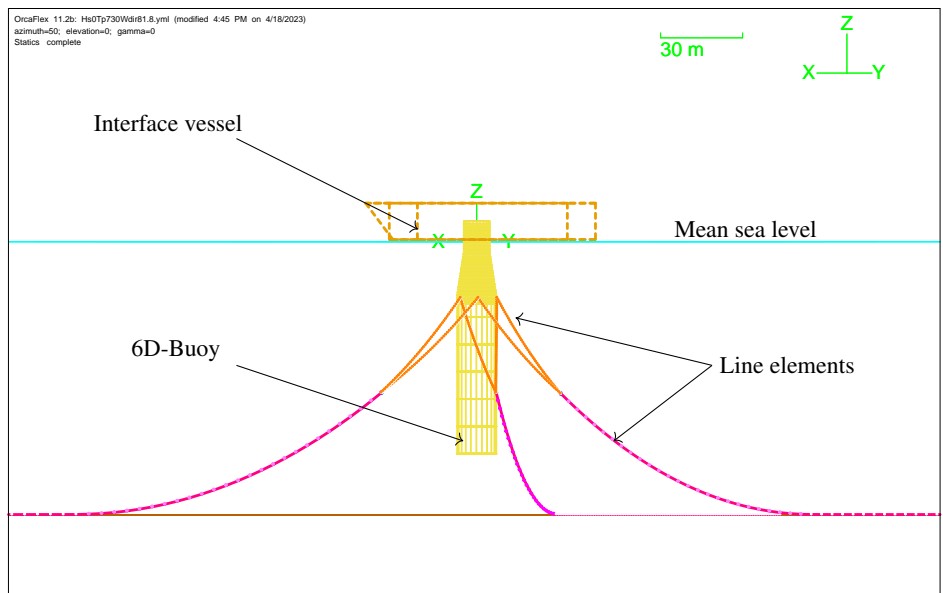

**Figure 3.** Schematic of the OrcaFlex simulation elements.

initial transients as the wave dynamics, turbulence and substructure motion build up as the artificial structural damping is slowly ramped down. The final post-processing is performed on 600s dynamic simulations that follow the initialization phase. The simulations for training the surrogate may be performed for a longer duration if necessary, mainly to estimate mooring line fatigue correctly due its long natural period. The effect on the tower and blade fatigue is shown to not change significantly with larger simulation windows, but rather with the fatigue calculation algorithm used to account for the unclosed cycles (Stewart

et al., 2013).

### 2.3    Definition of relevant site features and responses

Having a large feature space can lead to a very expensive surrogate training process, as the number of training samples needed grow with the number of input variables. It is, therefore, important to identify which variables have the largest impact on the fatigue. Several studies in the past have focused on addressing the sensitivity of wind turbine loads to environmental

conditions (Robertson et al., 2018, 2019; Teixeira et al., 2019; Shaler et al., 2023; Singh et al., 2024b). The combined effect of environmental and structural parameters has been analyzed on fixed-bottom (Hübler et al., 2017; Velarde et al., 2019) and floating wind turbines(Wang et al., 2023; Wiley et al., 2023; Lin et al., 2021; Reddy et al., 2024; Singh et al., 2024b).





Wiley et al. (Wiley et al., 2023) demonstrate that for the OC4-DeepCwind semi-submersible platform, the standard deviation of wind speed in the inflow is the most influential parameter affecting the fatigue and ultimate loads on the tower and blades. Secondary drivers of fatigue on the tower bottom moment include turbulence coherence parameters as well as wave characteristics, such as significant wave height and peak spectral period. For blade pitching fatigue, secondary factors include the yaw misalignment angle and geometric features like the blade twist angle. Whereas, the wind-wave misalignment and the current speed and direction seem to have a secondary effect on the blade root bending moment fatigue. Reddy et al. (Reddy et al., 2024) perform elementary effects analysis to determine the most significant parameters affecting tower bottom fatigue on the OC3-Hywind Spar platform and the OC4-DeepCwind semi-submersible design. In both cases, the significant wave height is found to be the primary driver. Current-related parameters are shown to have a strong effect mainly on the mooring line fatigue. Singh et al. (Singh et al., 2024b) use measurement data from the TetraSpar demonstrator to find that the tower and blade fatigue are most highly correlated with the wind speed mean and standard deviation, and significant wave height values.

As observed, although current has a big impact on the mooring line loads, its effect is found to not be as significant on the tower and blades in literature. Therefore, it is not included as a variable feature in the training of the surrogate. A variation in the water depth was also not considered because the mooring line system must be redesigned for any new value of water depth. A framework for automating this process is non-trivial. Furthermore, for slack mooring lines, Lin et al. (Lin et al., 2021) show a negligible impact on the tower fatigue with an increase in water depth.

**Table 4.** The list of input features provided to the surrogate model and their corresponding notation.

| Feature | Label |
|---|---|
| Wind speed [$\mathrm{ms}^{-1}$] | $U_{ref}$ |
| Shear exponent [$-$] | $\alpha$ |
| Turbulence Intensity [%] | $TI$ |
| Significant wave height [m] | $H_s$ |
| Spectral peak period [s] | $T_p$ |
| Wave direction [°] | $W_{dir}$ |
| Yaw error/ misalignment [°] | $Yaw$ |

**Table 5.** The list of output channels that the surrogate models are trained to predict.

| Response |
|---|
| Tower bottom fore-aft DEL [$-$] |
| Tower top fore-aft DEL [$-$] |
| Blade root edgewise DEL [$-$] |
| Blade root flapwise DEL [$-$] |

The targets the surrogates are trained on are listed in Table 5. Each target is trained with a separate surrogate model. This study only calculates the short-term DELs in the local coordinate system. DELs result from the conversion of the irregular load time series to a constant amplitude and frequency signal that produces an equivalent fatigue damage. Rainflow counting (Matsuishi and Endo, 1968) algorithm is used to obtain the load ranges $S_i$ and the number of load cycles $n_i$ needed to calculate the DEL as,

$$DEL := \left( \frac{n_i S_i^m}{n_{ref}} \right)^{1/m}, \tag{1}$$



where $n_{ref}$ is 600 for 1Hz DELs over 10 minutes. $m$ is the Wöhler coefficient with values 3.5 for the tower, 10 for blade flapwise, and 8 for blade edgewise moments.

## 2.4    Feature bounds

The feature bounds are defined based on the observations of data on sites where floating wind farms could potentially exist (Creane et al., 2024) and where data was readily available. Table A1 lists the selected sites with their location and water depth
values. The ERA5 reanalysis data, produced by the European Center for Medium-Range Weather Forecasts (ECMWF) on behalf of the European Union's Copernicus Climate Change Service (C3S), is used for the analysis in this section.

### 2.4.1    Average wind speed at hub height

The wind speed at hub height ($U_{ref}$) varies between 3 and $28\mathrm{ms}^{-1}$, which is the operational range of the wind turbine investigated in this study.

### 2.4.2    Shear exponent

The shear exponent $\alpha$ is defined according to the wind profile power law as,

$$\frac{U}{U_{ref}} = \left(\frac{z}{z_{ref}}\right)^{\alpha}, \tag{2}$$

where, $U$ is the wind speed at height $z$, and $U_{ref}$ is the known wind speed at height $z_{ref}$.

   We used the ERA5 reanalysis data to obtain wind speed values at 10m and 100m for the sites listed in Table A1. Assuming
the wind profile follows the power law, the shear exponent is calculated using Equation (2). The distribution of the shear exponent is shown in Figure 4a, with values primarily ranging between 0 and 0.2. It is also plotted against wind speed in Figure 4b. In our database, the shear exponent is uniformly distributed in the region corresponding to the dashed box in Figure 4b.

### 2.4.3    Turbulence intensity

The lower range of turbulence intensity is $0.1\%$, and the upper limit is designed to be $20\%$ greater than the prescribed IEC Class C standard (IEC, 2010) for the Normal Turbulence Model (NTM). The function for class C turbulence intensity (in percentage) is given by,

$$TI = 100 \times \frac{I_{ref}(0.75V_{hub} + 5.6)}{V_{hub}} \tag{3}$$

   where, $I_{ref} = 0.12$ is the expected value of turbulence intensity at $15\mathrm{ms}^{-1}$. The upper limit of turbulence intensity for
sampling is, therefore, $1.2 \times TI$. Figure 5 shows the chosen range, with the IEC class C turbulence and the measured turbulence at the FINO 3 metmast- which is referenced in the Buchanan deep met-ocean report (Equinor ASA, 2022).





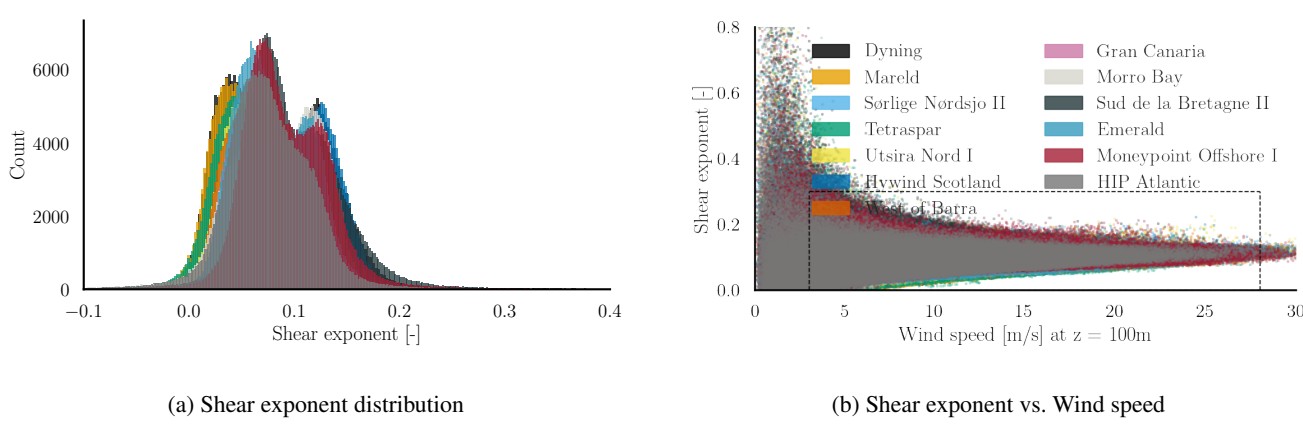

| (a) Shear exponent distribution | (b) Shear exponent vs. Wind speed |

**Figure 4.** (a) Histograms of the shear exponent $\alpha$ for selected sites for ERA5 reanalysis data from 1990 to 2019. (b) Shear exponent shown as a function of wind speed, marked by a box denoting the selected sampling domain.

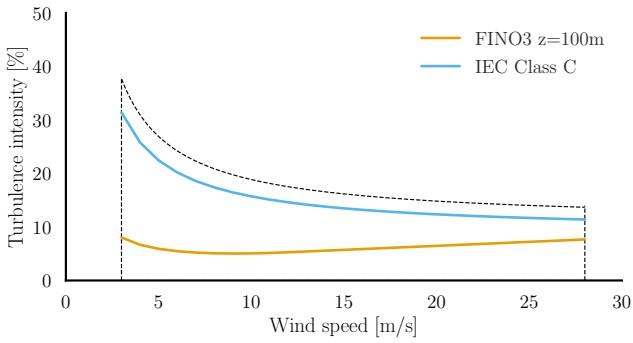

**Figure 5.** Chosen turbulence intensity range in dashed lines, along with the IEC class C turbulence profile for NTM, and measured turbulence at FINO 3 (German Bight) from the met-ocean analysis report on the Hywind Scotland project (Equinor ASA, 2022).

### 2.4.4 Significant wave height

Waves in deep water primarily originate from two sources: wind-induced waves and swell waves. It is useful to consider the correlation between wind speed and significant wave height while training the surrogate to avoid including non-physical wind-
205 wave combinations. Figure 6a illustrates a scatter plot of significant wave height ($H_s$) versus wind speed, based on ERA5 reanalysis data for the selected sites. The sampling domain is also a function of wind speed, highlighted with the dashed lines.

The upper and lower ranges of sampling for the significant wave height are defined empirically based on these observations. In this case, the functions are rather conservative and subject to modification based on the kind of sites the user would want to use the surrogate model on. The equations for the upper and lower limits for $H_s$ are listed in Appendix B1.





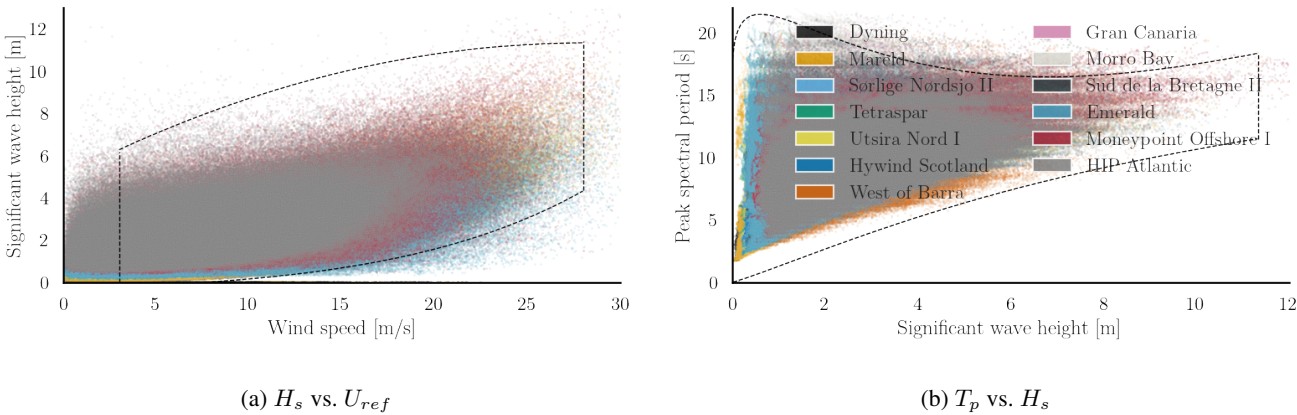

(a) $H_s$ vs. $U_{ref}$            (b) $T_p$ vs. $H_s$

**Figure 6.** Scatter plots of (a) the significant wave height vs. the wind speed at 100 m, and (b) peak spectral period vs. significant wave height for the selected sites (Table A1) based on the ERA5 reanalysis data from 1990 to 2019.

### 2.4.5 Peak spectral period

The empirical functions are defined for the spectral period range based on the significant wave height. Figure 6b illustrates the sampling domain with dashed lines, overlaid on observational data from the ERA5 reanalysis. This plot also includes $H_s - T_p$ values corresponding to wind speeds below the cut-in speed and above the cut-out speed. The functions defining this range are detailed in Appendix B2. As with the significant wave height, these bounding functions can be adjusted based on the region of primary interest to the user.

### 2.4.6 Wave direction

The wind turbine is always assumed to face the inflowing wind. Therefore, only the wave direction is varied to introduce wind-wave misalignment. Wave direction is considered to be an independent variable and sampled uniformly between $0°$ and $360°$. For asymmetric floating foundations, however, wind directions would also need to be considered as an independent parameter.

### 2.4.7 Initial yaw misalignment

The effect of the initial yaw misalignment is chosen to be evaluated at $-5.6°$, $0°$ and $5.6°$ while performing fatigue calculations. Therefore, we selected the sampling bounds between $1.1 \times -5.6°$ and $1.1 \times 5.6°$.

## 2.5 Training and testing database generation

### 2.5.1 Training database

Sobol sampling (Sobol, 1967) is used to jointly sample uniformly in seven dimensions to generate the training dataset. The samples lying outside the aforementioned feature bounds are discarded, resulting in a total of 9041 training samples. Each





sample corresponds to a unique wave seed in OrcaFlex and a single inflow turbulence seed in BHawC. This approach is designed to emulate the inherent stochasticity of real-world inflow variables. Note that the statistical variation in the flow field is constrained by the BHawC implementation to only 45 turbulence seeds. Consequently, these seeds had to be reused, and the inflow turbulence box could not be uniquely defined for every case.

### 2.5.2 Testing database

The values of the shear exponent, turbulence intensity and yaw misalignment are not randomly assigned to the test cases. Instead, they take the values used commonly while performing fatigue design load case evaluations. The shear exponent was fixed at $0.08$, with yaw misalignment values of $-5.6°$, $0°$, and $5.6°$, and turbulence intensity corresponding to IEC Class C values. $H_s$, $T_p$, $TI$, and $U_{ref}$ were jointly sampled in a random manner, without being tied to any specific location, but constrained within the defined feature bounds. In total, $n_{test} = 47$ test samples were used in this study. Each test sample simulation was repeated with $n_{seeds} = 44$ random seeds for turbulence and waves to capture the statistical variation in the DEL values from the variation in the wind and wave fields. The seed repetition establishes a *reference* conditional distribution for each sample, which is used to compare against the probabilistic predictions of the surrogate model in Section 4.2 . The samples used for training and testing the surrogate models are shown in Figure 7.

## 3 Methodology

This section briefly describes the theoretical basis of the mixture density network models investigated in this study, as well as the accuracy metrics considered to evaluate the surrogate's goodness of fit. The database $\{\boldsymbol{x}^q, y^q\}_{q=1\ldots n}$ consists of $n$ pairs of inputs $\boldsymbol{x} \in \mathbb{R}^d$, and the corresponding output $y \in \mathbb{R}$. The surrogate is calibrated separately for each target.

### 3.1 Mixture density networks

A mixture density network is a probabilistic regression method that combines Gaussian mixture models with artificial neural networks (Bishop, 1994). The conditional distribution of the target is represented as a linear combination of $m \in \mathbb{N}$ Gaussian kernel functions,

$$p(y \mid \boldsymbol{x}) = \sum_{i=1}^{m} \alpha_i(\boldsymbol{x}) \mathcal{N}(y \mid \mu_i(\boldsymbol{x}), \sigma_i^2(\boldsymbol{x})), \tag{4}$$

where $\alpha_i(\boldsymbol{x})$ are the weights or mixing coefficients assigned to the $i^{\text{th}}$ mixture component. $\mathcal{N}(y \mid \mu_i(\boldsymbol{x}), \sigma_i^2(\boldsymbol{x}))$ is a Gaussian kernel representing the conditional density of the $i^{\text{th}}$ component of the target distribution, with parameters $\mu_i(\boldsymbol{x})$ and $\sigma_i(\boldsymbol{x})$. Instead of mapping the inflow features $\boldsymbol{x}$ to the load statistics $y$ directly, the neural network is trained to predict the parameter vector, $\boldsymbol{z} \in \mathbb{R}$ consisting of $\alpha_i(\boldsymbol{x})$, $\mu_i(\boldsymbol{x})$ and $\sigma_i(\boldsymbol{x})$ for $1 < i < m$.

The mixing coefficients $\alpha_i(\boldsymbol{x})$ must sum up to exactly 1. A *softmax* function is used to handle this constraint. Positive values of the standard deviation are ensured by representing them as exponential functions of the corresponding network outputs, $z_i^\sigma$. The means are not constrained.



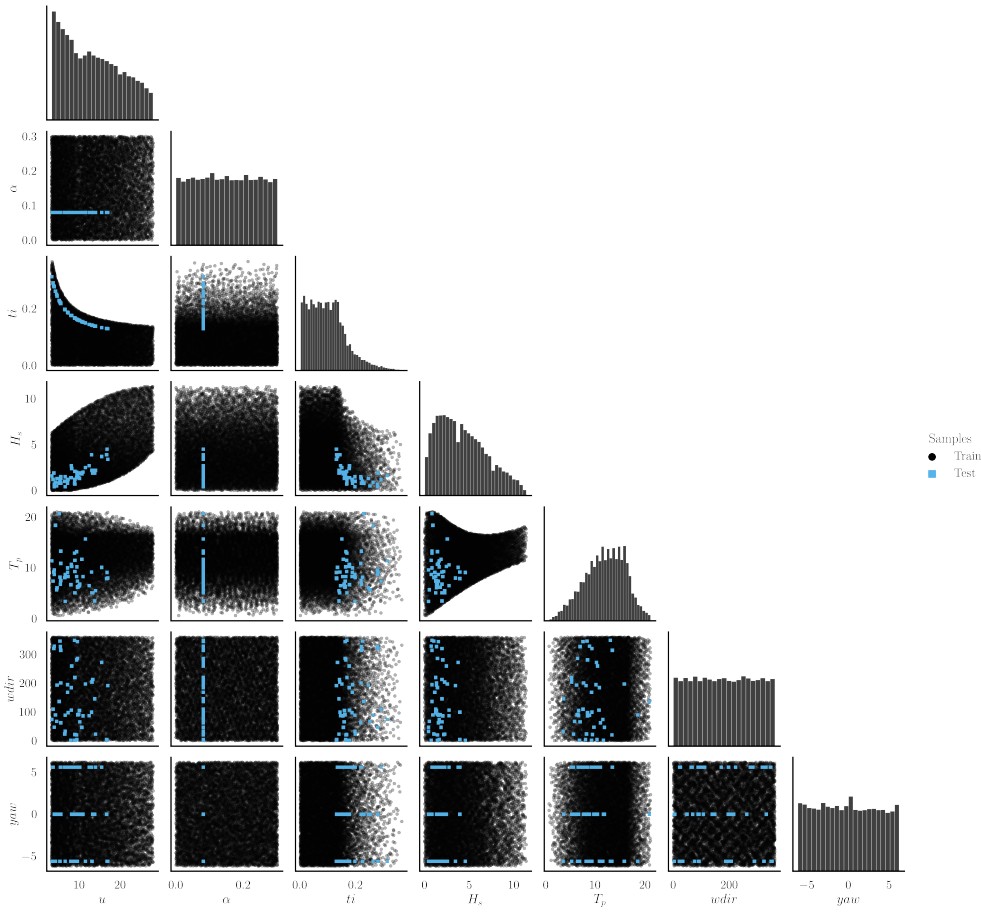

**Figure 7.** Paired scatter plots and marginal distributions of the training and testing datasets.

The error function $E^q$ is defined as the negative log of the likelihood. For pattern $q$, it is given by,

$$E^q = -\ln\left(\sum_{i=1}^{m} \alpha_i(\boldsymbol{x}^q)\mathcal{N}(y^q \mid \mu_i(\boldsymbol{x}^q), \sigma_i^2(\boldsymbol{x}^q))\right). \tag{5}$$

The likelihood of the dataset is the product of the likelihoods of the individual data samples.

The derivative of the error function is calculated at the output layer and is back-propagated to get its gradient with respect to the network weights. The values of the network parameters are adjusted to minimize the error function using a gradient descent optimization. This study uses the Adam optimizer (Kingma and Ba, 2017) to perform stochastic gradient descent. The model is initialized ten times for any given case in order to choose the best initial conditions for the optimizer. The hidden layers in our network use the rectified linear unit (ReLU). The output layer of the network does not have an activation function; therefore, 265    the outputs are just linear combinations of the inputs from the previous layer.





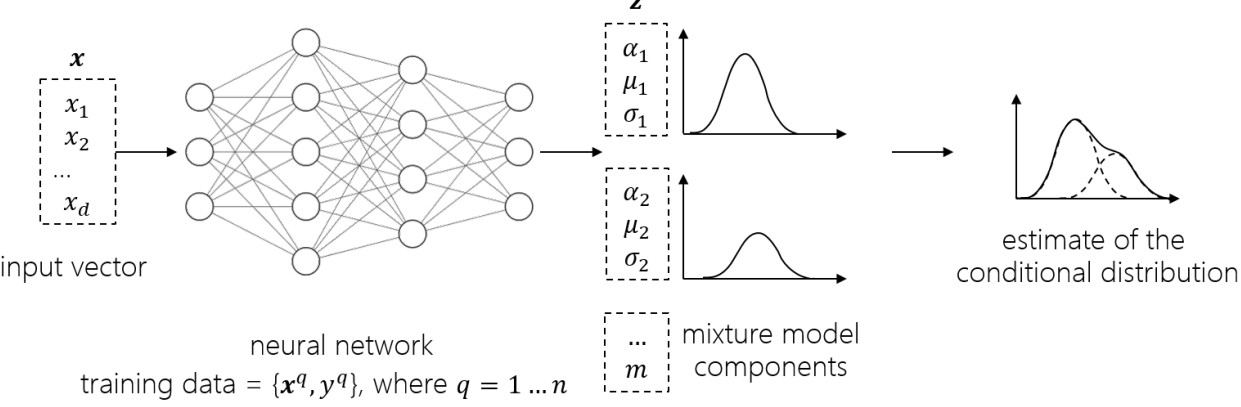

**Figure 8.** Schematic representation of Mixture Density Networks.

Minimizing the error function is an ill-posed problem as there is a conflict between learning the function that fits the data perfectly and remaining robust under varying sets of training data. As the network size grows, the function space increases and the neural network tends to overfit. the MDN model training especially seemed susceptible to it. Among several ways to avoid overfitting (Montavon et al., 2012), in this study, we implemented a combination of *early-stopping* (Yao et al., 2007) and *L1 and L2 regularization* (Ng, 2004).

The main hyperparameters used in this study to train the models to obtain the results in Section 4 are summarized in Table 6. In subsequent sections, we test the performance of the MDN model with various architectures. The features and targets are scaled with the standard scaler before training.

## 3.2 Accuracy metric

The qualitative assessment of the performance of the surrogate model is based on two criteria: the coefficient of determination ($R^2$) and the Wasserstein distance ($d_{W2}$), as described hereafter.

### 3.2.1 Coefficient of determination $R^2$

The coefficient of determination, also known as the $R^2$, is a common measure of the goodness of fit of a model. It is defined as,

$$R^2 = 1 - \frac{\sum (y_i - \hat{y}_i)}{\sum (y_i - \bar{y})}, \tag{6}$$

where $\hat{y}_i$ is the predicted output, $y_i$ is the observed value and $\bar{y}$ is the mean of the observed values. $R^2$ is interpreted as the linear correlation between the predicted and observed values of the output vector. To assess the accuracy of the predicted conditional distribution of the response compared to the BHawC reference, we calculate the $R^2$ value for the conditional probability density





**Table 6.** Summary of the network hyperparameters

| Network hyperparameter | Value |
| --- | --- |
| Number of mixture components | 4 |
| Activation function (hidden layers) | ReLU |
| Activation function (output layer) | None |
| Learning rate | $5e^{-3}$ |
| Maximum epochs | 5000 |
| Mini-batch size | 100 |
| Optimizer | Adam |
| Regularization | |
| $\lambda$ for $L1-$regularization | $1e^{-3}$ |
| $\lambda$ for $L2-$regularization | $1e^{-3}$ |
| Early-stopping | |
| Early-stopping patience | 100 |
| Early-stopping monitor | validation loss |
| Number of early-stopping validation samples | 600 |
| Steps per epoch | number of training samples / batch size |

function's (pdf's) mean and standard deviation. These two quantities are derived empirically by obtaining 5000 samples from

the surrogate-predicted conditional distribution and $n_{seeds}$ seed (turbulence and wave) repetitions per test case.

### 3.2.2   Wasserstein distance

The Wasserstein metric is a distance function that compares the difference between the pdfs of two random variables. It is symmetric, non-negative, and satisfies the triangle inequality, making it a proper distance metric. In the case of 1-D distributions, the Wasserstein-2 distance between a reference empirical measure $Y$ and predicted measure $\hat{Y}$, is defined as (Villani, 2009;

Peyré and Cuturi, 2019; Ramdas et al., 2015),

$$W_2(Y,\hat{Y}) = \left( \int_0^1 |F^{-1}(t) - G^{-1}(t)|^2 dt \right)^{1/2} \tag{7}$$

where $F^{-1}$ and $G^{-1}$ are the quantile functions of $Y$ and $\hat{Y}$ respectively. The individual quantile functions are obtained from the samples of the empirical distributions and then integrated. In this paper, we calculate the Wasserstein distance between the conditional distribution predicted for each sample ($\hat{Y}$) and the conditional distribution obtained as a reference through seed

repetitions in BHawC/OrcaFlex ($Y$). $\hat{Y}$ consists of 5000 samples from the surrogate's estimate, and $Y$ is obtained from $n_{seeds}$ turbulence and wave seed repetitions in BHawC/OrcaFlex. The distance metric is normalized by the standard deviation of the





reference conditional distribution, $Y$. Therefore, a value of $\frac{W_2}{\sigma(Y)} = 1$ is the distance between a distribution with mean $\mu(Y)$, scale $\sigma(Y)$, and a degenerate distribution with the same mean. We calculate the global performance of the model by averaging the normalized Wasserstein distance over $n_{test}$ test samples as,

$$300 \quad d_{W2} = \mathbb{E}_{n_{test}} \left( \frac{W_2}{\sigma(Y)} \right) \tag{8}$$

## 4 Results

This section is divided into three parts. The first part presents a convergence study on the number of training samples, high-lighting the model's robustness and demonstrating a clear trade-off between the computational cost of data generation and the resulting accuracy. A related hyperparameter study to determine the network architecture is presented in Appendix C. The second part validates the performance of the surrogate on the test dataset. The validated model is used to make lifetime fatigue damage estimates on the wind turbine components in response to different site conditions in the third section.

### 4.1 Choice of training data size

This section shows the convergence study with respect to the number of training samples for the tower bottom fore-aft DEL. It is assumed that the same architecture can be used to predict the remaining channels. Three networks for the mixture density networks are compared to test the robustness of the approach, as listed in Table 7. The MDNs contain four mixture elements. The rest of the hyperparameters are as specified in Table 6. In Figure 9, at each $N_{train}$ value, the models are trained on 25 different subsets of the total training data space to capture the sensitivity of the model's fit to the choice of the training samples. The boxes reflect the variation in the $R^2$ values as a result of the choice of training data points. The boxes extend between the data's first ($Q1$) and third ($Q3$) quartile, and the horizontal line across the box indicates the median. The difference between $Q1$ and $Q3$ defines the interquartile range ($IQR$). The upper whisker extends to the largest data values within $1.5IQR$ above $Q3$. The lower whisker, similarly, extends to the lowest data point within $1.5IQR$ below $Q1$. Outliers are visible as dots beyond the whisker boundaries. Figure 9a is the $R^2$ value obtained from predicting the mean of the conditional pdf of the tower bottom fore-aft DEL, averaged over the test dataset. The mean in the BHawC reference is calculated using 44 realizations of the wind and wave fields. The diminishing size of the $IQR$ as the number of samples grow is a combination of the increasing robustness of the model, and the smaller variability in the test samples as fewer untrained samples remain in the dataset.

**Table 7.** The MDN architectures considered for the convergence study.

| Notation | Number of layers | Number of nodes per layer |
|----------|------------------|---------------------------|
| MDN$[10, 10]$ | 2 | 10 |
| MDN$[30, 30]$ | 2 | 30 |
| MDN$[50, 50]$ | 2 | 50 |




MDN uses a neural network framework capable of inferring extremely complex underlying functions given sufficient data. In this case MDN coverges to consistent values of both $R^2$ of the conditional mean and $d_{W2}$ above 4250 samples. We also see that MDN estimates are closer to the ground truth with larger networks of 30 or 50 nodes per layer. The performance of MDN$[30,30]$ and MDN$[50,50]$ is almost identical in this region, indicating good model robustness with respect to the size of the layer.

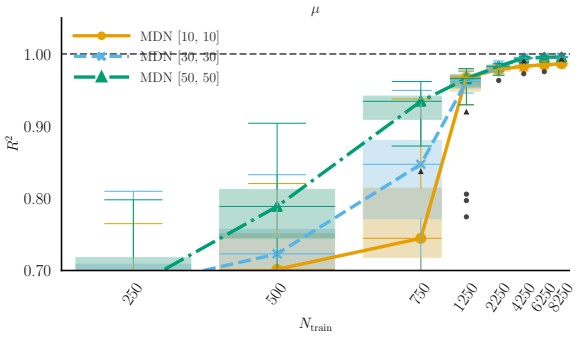
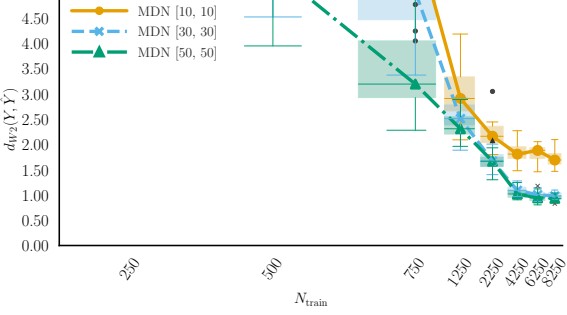

(a) $R^2$ values for the predicted mean

(b) $d_{W2}$ for the predicted conditional pdf

**Figure 9.** Convergence plots for the tower bottom fore-aft DEL channel. (a) Shows the convergence of the predicted mean as a function of the number of training samples for three MDN architectures. (b) Shows the normalized Wasserstein distance as a function of the training samples.

Figure 9b shows the normalized 2-Wasserstein distance between the predicted and reference pdf. The Wasserstein distance quantifies the similarities between the predictions and the reference. It is, thus a good indicator of whether or not the surrogate can correctly estimate the variation in the target resulting from a combination of epistemic and aleatoric sources. The predicted pdf is based on 5000 realizations from the estimated Gaussian mixture in MDN. The reference is based on $n_{seeds} = 44$ BHawC/OrcaFlex realizations. Since 44 samples are insufficient to characterize the reference pdf fully, there is certainly an error associated with the $d_{W2}$ values; therefore, $d_{W2}$ cannot be expected to be zero in practice. Beyond 4250 samples, there is a small but marginal improvement in the $d_{W2}$ values from MDN$[30, 30]$ and MDN$[50, 50]$.

In conclusion, the two-layered MDN surrogates (MDN$[30, 30]$ and MDN$[50, 50]$) reach convergence in terms of $d_{W2}$ at 4250 samples. For subsequent sections, the MDN models will be trained with a dataset of 8250 points, as this provides marginally better predictions with only a slight increase in model fitting cost.

The choice of the number of layers and nodes in the neural network, as well as the number of mixture components is based on a hyperparameter study, presented in Appendix C.





## 4.2 Surrogate model validation

In this section, the performance of the MDN model is evaluated for the load channels listed in Table 5, on the selected test
dataset presented in Section 2.5.2. Based on studies in Section 4.1 and Appendix C, the values of hyperparameters used in
training the models, in addition to Table 6, are listed in Table 8.

**Table 8.** List of hyperparameters used for training the MDN model for the final load prediction.

| MDN hyperparameter | Value |
|---|---|
| Number of hidden layers | 3 |
| Width layer 1 | 30 |
| Width layer 2 | 30 |
| Width layer 3 | 50 |
| Number of mixture components | 4 |
| Number of training samples | 8250 |

Table 9 provides a quantitative analysis of the model's performance in terms of the average $R^2$ and $d_{W2}$ values. The con-
ditional mean is accurately captured by the MDN model with $R^2$ exceeding 0.99 on the test dataset. The goodness of fit on
the conditional distribution is evaluated using $d_{W2}$. Lower $d_{W2}$ values indicate a smaller difference between the predicted and
reference conditional distributions across the test database. As the $d_{W2}$ values are normalized by the local reference standard
deviation, we can compare the performance of the models across different load channels. MDN's performance remains consis-
tently good on the tower top and blade targets. The tower bottom channel shows a larger deviation in the $d_{W2}$ values, which is
investigated further in Figure 10.

**Table 9.** Quantitative analysis of MDN model's predictions using $d_{W2}$ and $R^2$ as evaluation metrics.

| Model | Tower bottom FA | Tower top FA | Blade root edgewise | Blade root flapwise |
|---|---|---|---|---|
| $d_{W2}$ | 0.86 | 0.35 | 0.36 | 0.36 |
| $R^2\mu$ | 0.99 | 0.99 | 0.99 | 0.99 |

Figure 10 shows the statistics of the conditional distribution of the DEL variation at the tower bottom fore-aft direction Since
44 seeds is a relatively small sample size to determine the true mean and standard deviation of the population, a gray area is
highlighted in Figure 10a and Figure 10b to reflect the uncertainty in the reference values. For the mean, the 95% confidence
interval ($CI_t$) is calculated with the t-distribution (Rouaud, 2013), assuming the response is normal. It is defined as,

$$CI_t = \mu_{reference} \pm t.\frac{\sigma_{reference}}{\sqrt{n_{seeds}}}, \tag{9}$$

where $\mu_{reference}$ is the mean and $\sigma_{reference}$ is the standard deviation calculated from the simulation samples. $n_{seeds} = 44$ is
355 the number of seeds with which the simulations were repeated. $t$ is the *t-score* for 95% confidence, given $n_{seeds}$ samples from





a normally distributed population. The bounds are similarly calculated using the $\chi^2$ distribution for the standard deviation. The bounds are asymmetric as the $\chi^2$ distribution is skewed. $\chi_L$ and $\chi_R$ are based on $5\%$ and $95\%$ tails of the $\chi^2$ distribution. The true standard deviation, $\sigma$, is expected to lie between the bounds,

$$\sqrt{\frac{(n_{seeds}-1)}{\chi_L^2}}\sigma_{reference} \leq \sigma \leq \sqrt{\frac{(n_{seeds}-1)}{\chi_R^2}}\sigma_{reference}. \tag{10}$$

The individual pdfs are shown in Figure 10 for two site conditions. The reference BHawC realizations are plotted as histograms overlayed with kernel density estimate (KDE) plots generated from 5000 samples from the conditional pdf predicted by the surrogate model. The estimates include two sources of uncertainty. The first is the epistemic uncertainty of inferring a function from limited data. The second is due to the irreducible noise term, which is a part of the observed stochastic process. The subsequent plots assume the standard deviation of the combined uncertainty.

Figure 10a shows the predicted conditional mean ($\mu_{surrogate}$) of the normalized tower bottom fore-aft DEL as a function of the reference conditional mean ($\mu_{reference}$) derived from BHawC/Orcaflex simulations. As already indicated in Table 9, the $R^2$ values are greater than 0.99 for MDN, indicating an excellent fit. Similarly, the standard deviation derived from the surrogates ($\sigma_{surrogate}$) is plotted against the ground truth reference ($\sigma_{reference}$) in Figure 10b. Despite the slight overprediction of the standard deviation, MDN is able to capture the heteroscedastic trend in the data. Figure 10c corresponds to a below-rated

velocity of $9.8\mathrm{ms}^{-1}$ and a wind-wave misalignment of $105°$. Under these conditions, the BHawC reference is a short-tailed conditional pdf. Since MDN assumes a medium-tailed Gaussian mixture conditional, there is a tendency for the surrogate model to overestimate the standard deviation (Figure 10b). The reason for the tower bottom fore-aft DELs to be restricted between a very small range resulting in such a short-tailed distribution is not obvious and demands a deeper investigation into the behavior of the tower structure and control laws, which is beyond the scope of this paper. A similar pattern is not observed

in the other three load channels. Figure 10d corresponds to an example of a near-rated wind speed case, where the MDN predictions show a closer match to the reference conditional distribution.

A similar analysis is performed for the tower top fore-aft DEL channel in Figure 11. The conditional standard deviation estimated from the MDN surrogate are within the error bounds of the small population assumption, indicating a very good fit.

Figure 12 and Figure 13 show the surrogate models' performance on the blade root flapwise and edgewise DEL respectively.

Similar to the tower top, the standard deviation and mean estimates from the MDN surrogate agree very well with the BHawC reference in both blade channels.

These results indicate that the surrogate model demonstrates a good level of reliability in accurately predicting the DELs with respect to the BHawC/OrcaFlex reference. Consequently, we assume that the model can be extended to other operating conditions without necessitating further verification.

## 4.3    Lifetime damage equivalent loads

The calculation of aggregated fatigue loads in onshore wind cases consists of binning the wind speed and scaling the loads at each bin by the probability of occurrence of the wind speed during the operating lifetime of the wind turbine. Floating wind





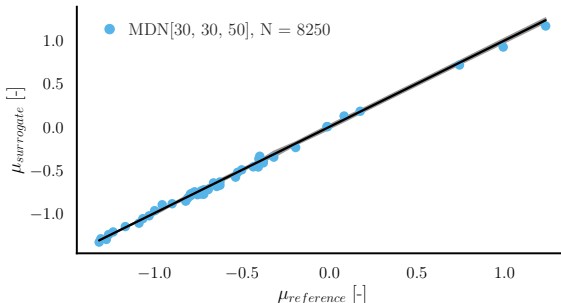

(a) Prediction of the mean DEL

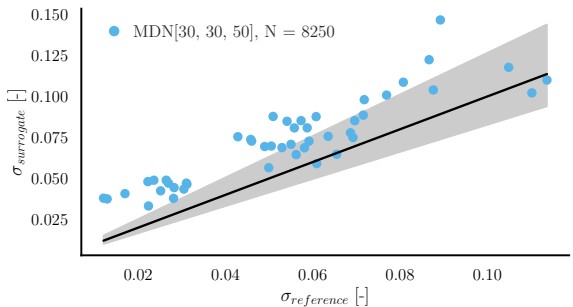

(b) Prediction of the standard deviation of the DEL

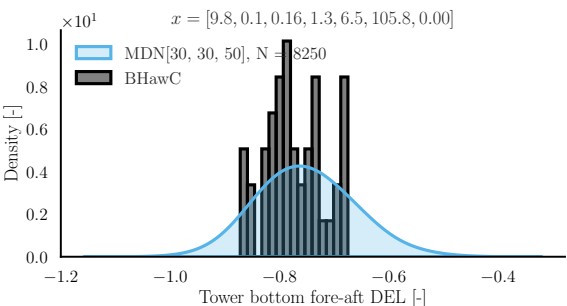

(c) Conditional pdf below rated conditions

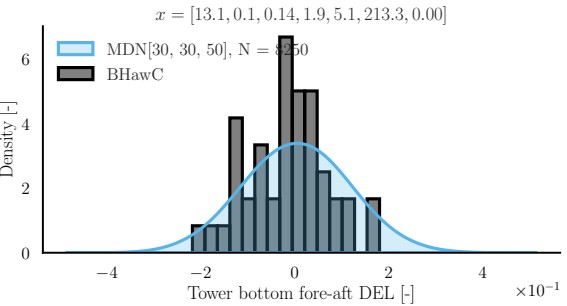

(d) Conditional pdf near rated conditions

**Figure 10.** Load predictions using MDN for the tower bottom fore-aft DEL (normalized). Figure (a) shows the surrogate predicted conditional mean at the test locations vs. the conditional mean calculated using BHawC. Figure (b) shows the predicted and reference standard deviations of the conditional pdf. Figures (c) and (d) compare the conditional pdf plots between the surrogate and the simulation at below rated and near rated conditions respectively. The values in vector $x$ denote: $[U_{ref}, \alpha, TI, H_s, T_p, W_{dir}, Yaw]$ with units specified in Table 4.

turbine fatigue evaluations are more complex, firstly, many more environmental parameters must be considered to characterize the site. Secondly, the bins need to be defined on a joint probability space. The process of choosing the right variables for the fatigue analysis and the size of the bins is not yet standardized and is a topic of ongoing research (Papi and Bianchini, 2024). With a fast surrogate model, however, it is possible to account for every single observation in the previous years, without the need to lump probabilities or limit the number of variables. The joint probabilities of the sea states are, therefore, automatically accounted for.

In this section, we use the validated surrogate model from Section 4.2 to make probabilistic estimates of the equivalent loads ($M_{eq}$) for 10 million reference load cycles on the floating wind turbine structure. The site data is obtained from the ERA5 database for four sites with an approximate water depth of 100m, namely Sud de la Bretagne II, Emerald, Hywind Scotland and HIP Atlantic (Table A1). Figure 14 provides an overview of the site conditions observed at the four selected sites. For simplicity, the foundation and mooring line design are assumed to be the same across the four sites. It is assumed that the





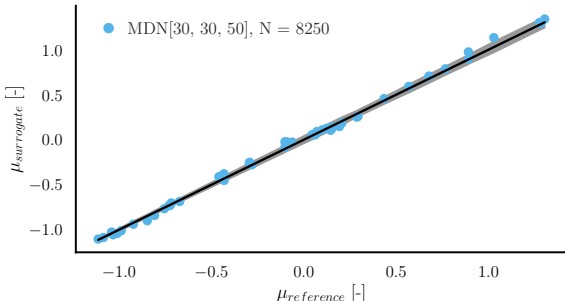

(a) Prediction of the mean DEL

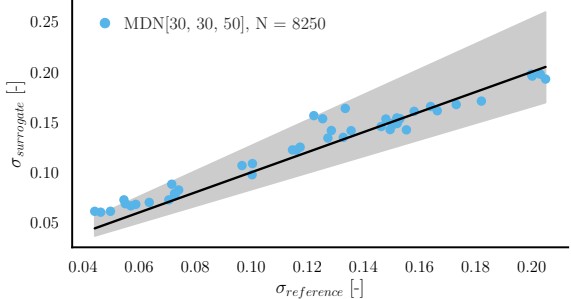

(b) Prediction of the standard deviation of the DEL

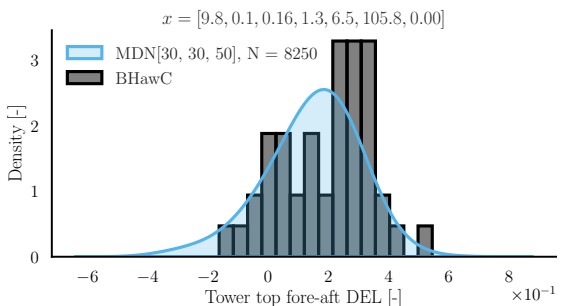

(c) Conditional pdf below rated conditions

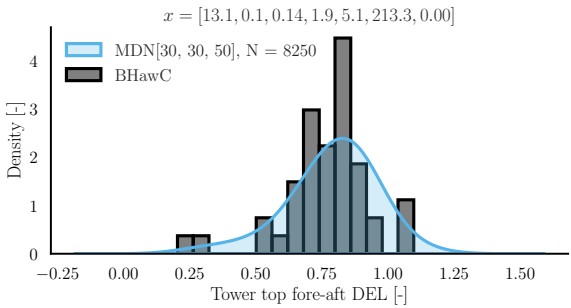

(d) Conditional pdf near rated conditions

**Figure 11.** Load predictions using MDN for the tower top fore-aft DEL (normalized). Figure (a) shows the surrogate predicted conditional mean at the test locations vs. the conditional mean calculated using BHawC. Figure (b) shows the predicted and reference standard deviations of the conditional pdf. Figures (c) and (d) compare the conditional pdf plots between the surrogate and the simulation at below rated and near rated conditions respectively. The values in vector $x$ denote: $[U_{ref}, \alpha, TI, H_s, T_p, W_{dir}, Yaw]$ with units specified in Table 4.

difference in the load distributions between the design in use and the site-optimized foundation will not be significant. The ERA5 hourly conditions are converted to 10-minute inputs by repeating each set of values six times. An alternative approach could be to draw the 10-minute values from a normal distribution with the hourly values a the mean and an assumed standard deviation. The observations below cut-in and above the cut-out wind speed are excluded from the calculations. The site data consists of the average wind speed at 100m, significant wave height, peak spectral period, wind-wave misalignment (converted to wave direction in OrcaFlex coordinates) and the shear exponent. The yaw misalignment values are sampled from a normal distribution with zero mean and a standard deviation of $2°$. The turbulence intensity is calculated for each case based on the wind speed assuming the IEC 61400-1 turbulence class C classification.

The value $M_{eq}$, represents the cyclic load amplitude which produces the equivalent lifetime damage given $n_{eq}$ cycles of oscillation over $L = 25$ years. In Equation (11), $M_i$ is the DEL for the $i^{th}$ 10-minutes of operation, $n_{ref}$ is the reference number of cycles per 10-minutes, set to 600. $m$ is the Wöhler coefficient, with part-specific values listed in Section 2.3. $n_L$





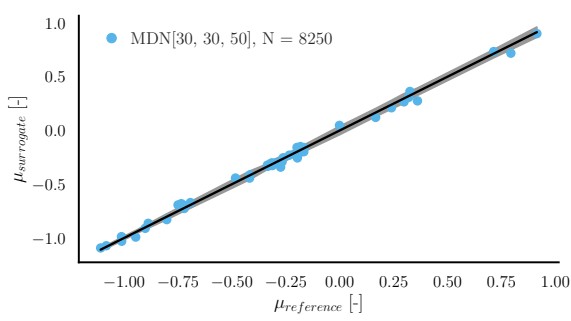

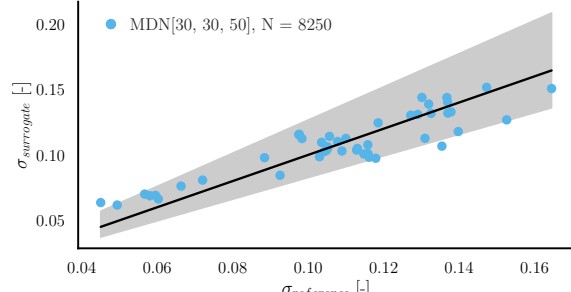

(a) Prediction of the mean DEL (b) Prediction of the standard deviation of the DEL

**Figure 12.** Load predictions using MDN for the blade root flapwise DEL (normalized). Figure (a) shows the surrogate predicted conditional mean at the test locations vs. the conditional mean calculated using BHawC. Figure (b) shows the predicted and reference standard deviations of the conditional pdf.

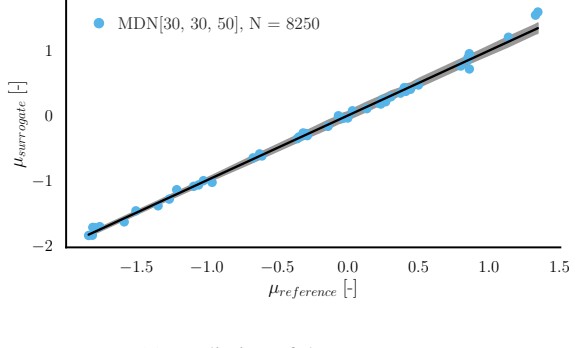

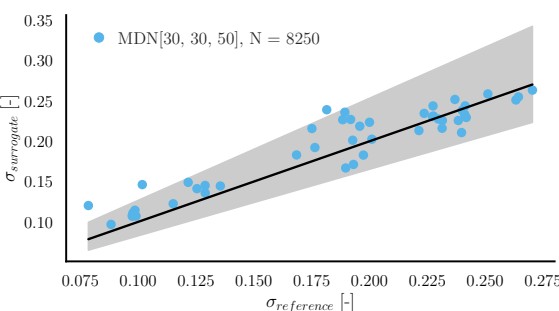

(a) Prediction of the mean DEL (b) Prediction of the standard deviation of the DEL

**Figure 13.** Load predictions using MDN for the blade root edgewise DEL (normalized). Figure (a) shows the surrogate predicted conditional mean at the test locations vs. the conditional mean calculated using BHawC. Figure (b) shows the predicted and reference standard deviations of the conditional pdf.

is the number of 10-minute periods in $L$ years. The loads do not have to be scaled as the probability of occurrence of each condition is equal. Since the surrogate has been validated in previous sections, we assume here that its predictions are accurate, and we can treat each $M_i$ as a probabilistic output from the MDN model. From each $M_i$ pdf, we draw 500 samples, resulting in a probabilistic estimation of $M_{eq}$. $M_{eq}$ is defined as,

$$M_{eq} = \left( \frac{n_{ref}}{n_{eq}} \sum^{n_L} M_i^m \right)^{1/m} \tag{11}$$





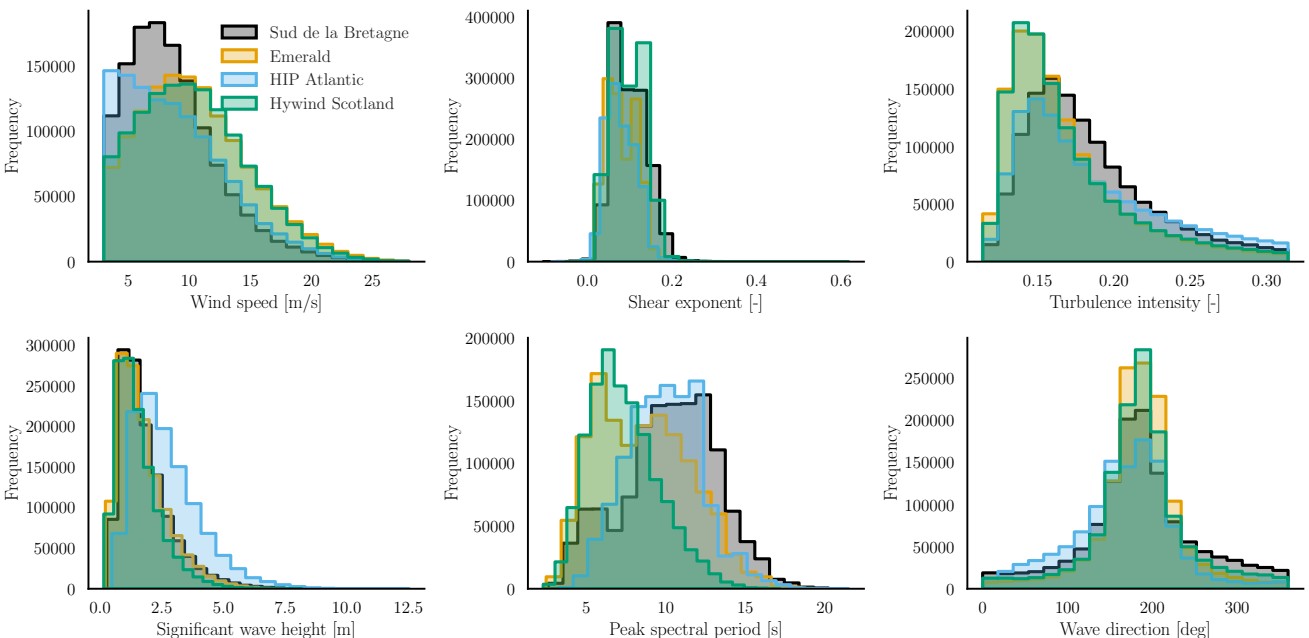

**Figure 14.** Comparison of the site conditions at the four floating wind sites considered in this study (Table A1).

where $n_{eq}$ is $10^6$ and $n_{ref}$ is fixed to 600 oscillations per 10 minute period. The probabilistic $M_{eq}$ value can be further used to calculate the stress reserve factor when re-designing the tower, or to calculate the fatigue damage during the structure's operating lifetime.

Figure 15 shows the kernel density estimate of the normalized $M_{eq}$ values from the surrogate for the four selected sites. $M_{eq}$ has been normalized by the average of the predicted $M_{eq}$ values at the Hywind Scotland site for every channel. Firstly, it is interesting to note that the uncertainty in $M_{eq}$ at each site is very small compared to the mean. This aligns with the law of large numbers, which states that for $M_i$ with a mean $\mu$ and variance $\sigma^2$, the standard deviation of the average of the distribution of $(\sum M_i)$ decreases as $\sigma/\sqrt{n_L}$. Since $n_L$ is in the order of $10^6$, the standard deviation becomes extremely small as we get closer to the true mean. Even though the effect of $M_i$ being raised to the power of $m$ means that any variability in the sum is amplified, subsequently taking the $m^{th}$ root has the opposite, damping effect. Therefore, the effect of the outliers is essentially nullified due to the averaging. It is important to note that this study considers only the statistical uncertainty arising from stochastic input sources. In practice, other sources of uncertainty may contribute to the analysis (IEC, 2024b). For example, uncertainties related to the underlying joint distribution of site conditions represent another significant source of variability. Including these additional uncertainties in the feature set would likely increase the variance of the final load estimates.

Secondly, the loads on different channels do not scale uniformly across sites. At the HIP Atlantic site, for instance, the cumulative tower bottom fore-aft moment is the highest, as shown in Figure 15. This is primarily due to the influence of the significant wave height, which is expected to have a larger impact on the tower bottom fatigue (Singh et al., 2024b; Wiley



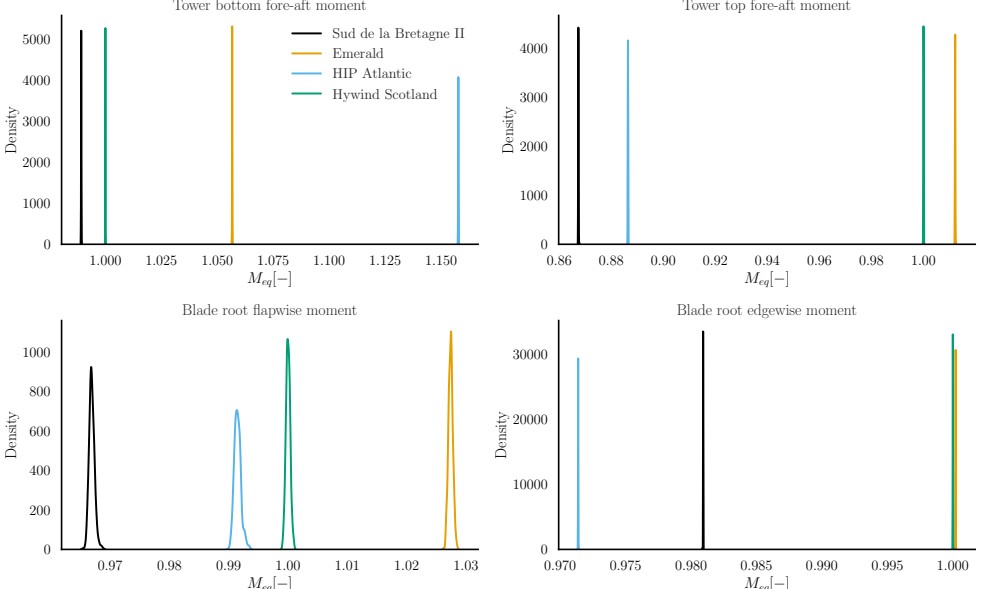

**Figure 15.** The 25-year normalized $M_{eq}$ calculated for four sites at the tower bottom fore-aft direction (top-left), tower top fore-aft (top-right), blade root flapwise (bottom-left) and blade root edgewise (bottom-right) channels. The mean $M_{eq}$ obtained at the Hywind Scotland site is used as the reference to normalize the loads at the remaining locations.

et al., 2023; Edwards et al., 2023). The marginal distribution of significant wave height at this site shows a higher probability of larger waves compared to other locations, supporting the observed increase in tower bottom loads.

The distributions of wind speed, turbulence intensity, and significant wave height at the Emerald and Hywind Scotland sites
(Figure 14) are nearly identical. This results in comparable tower top fore-aft and blade root edgewise damage. However, there remains a significant difference in blade root flapwise fatigue accumulation. This result is surprising, given that the loads at this location are primarily wind-driven. Nevertheless, it underscores the complexity of fatigue damage accumulation, which can yield different outcomes with minor variations in site conditions even with respect to non-dominant variables.

## 5 Conclusions

This paper presents a framework to develop probabilistic surrogate models for predicting floating offshore wind turbine fatigue loads for site analysis. The surrogate maps the environmental conditions from potential farm sites to the 10-minute damage equivalent loads experienced by a spar-type floating wind turbine. The main advantage of using probabilistic surrogates for this application is the ability to estimate conditional statistics with high accuracy to account for the statistical uncertainty resulting from the stochastic site conditions while minimizing the computational cost of training by avoiding seed repetitions. Based on





the reanalysis data from the ERA5 database for several comparable floating sites, the surrogate model is used to propagate the statistical uncertainties to the 25-year fatigue loads on the wind turbine.

In this study, the analysis is performed on a spar-buoy floating foundation based on a modified Hywind-Scotland 6MW wind turbine. The damage equivalent loads are considered on critical locations on the tower and blades and are calculated using a coupled implementation of BHawC/OrcaFlex for training and validating the surrogate. The features characterizing a
450 floating farm site and the appropriate ranges are defined. The probabilistic model considered in this study is the mixture density network, as it is flexible, robust, and interpretable and has performed well for fixed bottom load emulation in the literature.

Since MDN is based on a neural network parametrization, several hyperparameters require tuning prior to training. Therefore, a hyperparameter study is performed to find the appropriate neural network layout and the number of training samples to maximize the prediction accuracy of the MDN surrogate model. The conditional distribution predicted by the chosen model is
455 validated on a set of 47 operating conditions, each simulated with 44 random seeds in BHawC/Orcaflex to obtain a reference conditional distribution for each test case. The $R^2$ value for estimating the conditional mean is $> 0.99$ on all channels with the surrogate, indicating an excellent fit. The standard deviation of the conditional distribution is over-predicted by the model in the case of the tower bottom fore-aft moment but within the range of uncertainty bounds for the tower top and blade root channels.

Finally, the validated surrogate model is used to make probabilistic estimates of the 25-year equivalent damage on the tower and blades for four different sites. Since the surrogate model is fast, load predictions can be made quickly on all observed site conditions without lumping or binning the sea states a priori. The uncertainty in the aggregated lifetime fatigue loads due to statistical variance in the inputs is found to be much smaller in scale compared to the mean. This results from summing the 10-minute DELs over a million occurrences, effectively nullifying the impact of the outliers. We demonstrate that surrogate
models can be powerful tools for site analysis, especially for floating wind turbines, where the choice of variables and binning methods is still an open question. Additionally, using probabilistic surrogates like MDNs helps reduce bias in calculating the aggregate mean fatigue, as the conditional distributions are not always normally distributed.

Future studies could use such surrogates to identify optimal methods for grouping sea states in order to reduce the number of physics-based simulations required to achieve the same lifetime fatigue loads as using all observed site data. This type of
470 analysis would be computationally impractical with an engineering tool, as it would require performing millions of simulations to establish a baseline reference. Surrogates offer an alternative for reducing the computational demands while maintaining accuracy. Surrogate models can also be used in this context to isolate combinations of sea states that produce the highest fatigue on the wind turbine structure. Furthermore, it is interesting to include other sources of uncertainty in the analysis of loads. Once trained, probabilistic surrogate models can be used to propagate the different uncertainty sources to the loads to
475 study the combined effect without additional costs. This approach opens new opportunities for integrating reliability-based decision-making into the design process.





**Table A1.** Description of the sites used for defining the feature ranges.

| Site | Location | | ERA5 approx. location | | |
|------|----------|--|-----------------------|--|--|
| | Latitude [°] | Longitude [°] | Latitude [°] | Longitude [°] | Depth [m] |
| Dyning (Creane et al., 2024) | 58.218 | 17.860 | 58.00 | 17.75 | 141 |
| Mareld (Creane et al., 2024) | 58.161 | 10.575 | 58.25 | 10.50 | 233 |
| Sørlige Nørdsjo Phase II (Creane et al., 2024) | 56.783 | 4.918 | 56.75 | 5.00 | 60 |
| Tetraspar | 59.15 | 5.013 | 59.00 | 5.00 | 200 |
| Utsira Nord Phase I (Creane et al., 2024) | 59.276 | 4.540 | 59.00 | 4.50 | 273 |
| Buchanan Deep (Equinor ASA, 2014) | 57.45 | −1.31 | 57.50 | −1.25 | 100 |
| West of Barra (Vigara et al., 2019) | 56.885 | −7.947 | 57.00 | −7.75 | 100 |
| Gran Canaria (Vigara et al., 2019) | 27.75 | −15.33 | 27.75 | −15.00 | 200 |
| Morro Bay (Vigara et al., 2019) | 35.083 | −121.5 | 35.5 | −121.75 | 870 |
| Sud de la Bretagne II (Creane et al., 2024) | 47.3247 | -3.6594 | 47 | -3.7 | 94 |
| Emerald (Creane et al., 2024; Wind, 2025) | 51.3565 | -8.0761 | 51.5 | -8 | 90 |
| Moneypoint Offshore I (Creane et al., 2024; ESB) | 52.519 | -10.276 | 52.5 | -10.5 | 102 |
| HIP Atlantic (Creane et al., 2024) | 63.6325 | -16.3756 | 63.5 | -16.5 | 98 |

## Appendix A: ERA5 locations used for defining feature ranges

Table A1 lists the locations used for defining the feature ranges in Section 2. The data is downloaded from the years 1979 to 2020. The database consists of the hourly average wind speeds at $10\,\mathrm{m}$ and $100\,\mathrm{m}$, the significant wave height, spectral peak
period, and wave direction. The shear law exponent is derived from the wind speed values assuming a power law profile for the atmospheric boundary layer.

## Appendix B: Feature bounds

### B1 Significant wave height

The upper and lower limits for the significant wave height are defined as functions of the wind speed at hub height $U_{ref}$.
The upper limit is a quadratic function of the form:

$$H_{s_U} = -0.008U_{ref}^2 + 0.45U_{ref} + 5 \tag{B1}$$

The lower limit is defined as:

$$H_{s_L} = 0.719e^{(0.0832U_{ref})} - e^{(0.04U_{ref})} \tag{B2}$$





## B2    Peak spectral period

The peak spectral period range is designed to be a function of the significant wave height (which is in turn, a function of the wind speed at hub height). We define scaling functions $A$, $B$ and $C$ as,

$$A = a_1 + a_2 H_s^{a_3} \tag{B3}$$

$$B = b_1 + b_2 e^{-b_3 H_s} \tag{B4}$$

$$C = c_1 + c_2 e^{-c_3 H_s} \tag{B5}$$

The scaling functions are used to define the upper bound $T_{p_U}$ and lower bound $T_{p_L}$ as,

$$T_{p_\mu} = e^{(A + 0.5B)} \tag{B6}$$

$$T_{p_L} = T_{p_\mu}(1 - 3 \times \sqrt{(e^B - 1)}) \tag{B7}$$

$$T_{p_U} = T_{p_\mu}(1 + 3 \times \sqrt{(e^C - 1)}) \tag{B8}$$

The coefficients used to fit the curve in this study are listed in Table B1.

| $a_1$ | $a_2$ | $a_3$ | $b_1$ | $b_2$ | $b_3$ | $c_1$ | $c_2$ | $c_3$ |
|---|---|---|---|---|---|---|---|---|
| 1.3 | 0.57 | 0.37 | 0.005 | 0.1 | 0.43 | 0.005 | 0.75 | 0.6 |

**Table B1.** Tuning coefficients for defining the range functions for the spectral wave period.

## Appendix C:  Choice of hyperparameters

### C1    Number of layers and nodes

Large networks are better at capturing complex expressions in data but are susceptible to overfitting with a small training set. The objective of this study is first, to observe the robustness of the model relative to the number of network parameters for a particular training data size. And second, to choose a network architecture suitable for the rest of the study.

A sensitivity study on the number of nodes and layers is performed in this section for a training dataset of 8250 samples. The number of mixture parameters is 4 in all cases, and the rest of the network hyperparameters are fixed to the values listed in Table 6. Networks with 2, 3, and 4 layers with various widths are tested. The x-axis in Figure C1 lists the combinations of widths per layer evaluated in this study. The tower bottom fore-aft DEL channel is chosen for this study.

The $R^2$ values of the DEL are notably good for the architectures tested, indicating good model robustness. The main dif-

ferences observed are in $d_{W2}$, where the large 3 or 4-layer networks are generally better at capturing the complete conditional



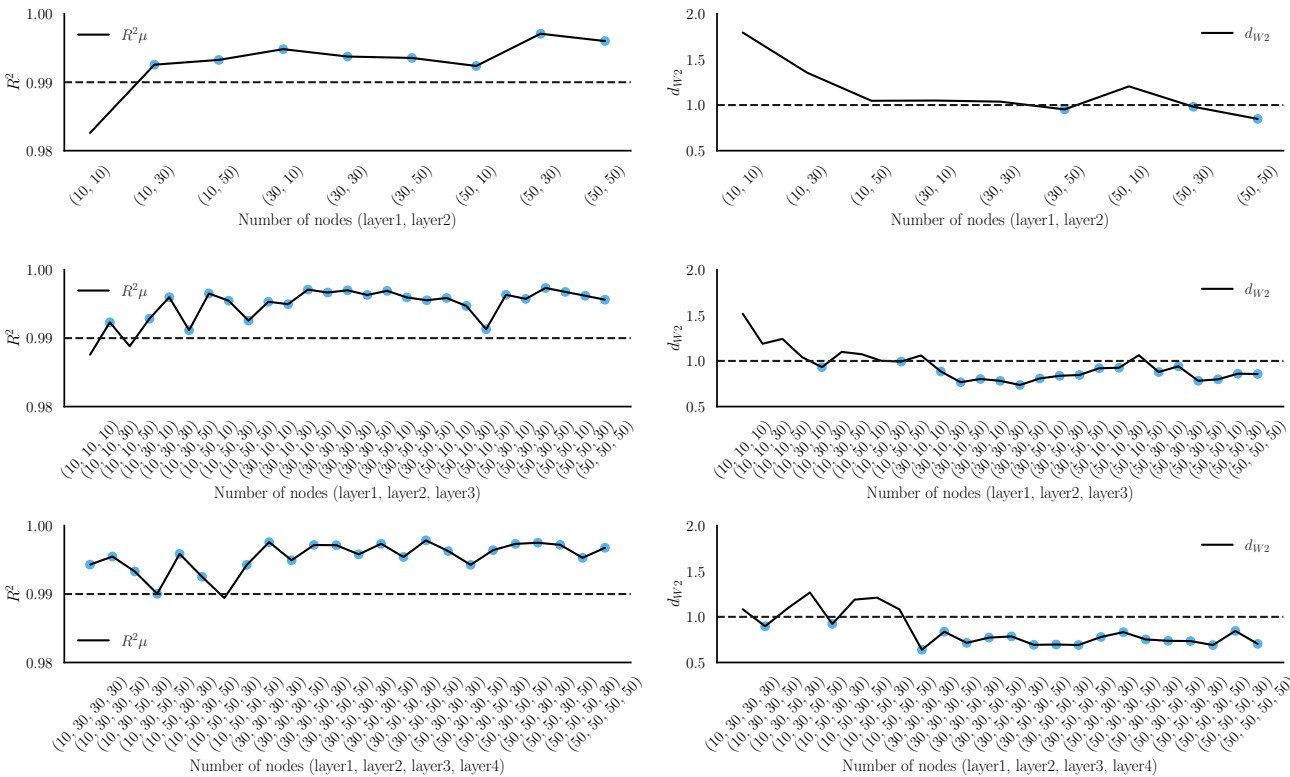

**Figure C1.** Study on the network architecture. The x-axis reflects the number of nodes per layer. The rows correspond to 2, 3, and 4-layer networks. The left column shows the $R^2$ value for the mean of the conditional pdf of the tower bottom fore-aft DEL channel. The dashed line corresponds to an $R^2$ value of 0.99. The right column plots the $d_{W2}$ values for the same channel with the dashed line corresponding to $d_{W2}$ of 1.

pdf. For the remainder of this study, we chose a 3-layer network with 30 nodes in layer 1, 30 nodes in layer 2, and 50 nodes in layer 3 in combination with 8250 training samples.

## C2  Number of mixture elements

The number of Gaussian distributions in the mixture controls the complexity of the predicted conditional pdf. However, a large
number of unnecessary mixture elements add redundancy and increase the computational complexity of the surrogate. In this section, we use 6250 and 8250 training samples with a 3-layer architecture width = (30, 30, 50) and test the performance of 4, 12, and 20 mixture elements on the tower bottom fore-aft DEL channel.

The number of components does not affect the estimation of the tower bottom fore-aft DEL mean. A slight improvement can be seen in Figure C2b with four components. The model, therefore, appears to be robust regarding the choice of the number of





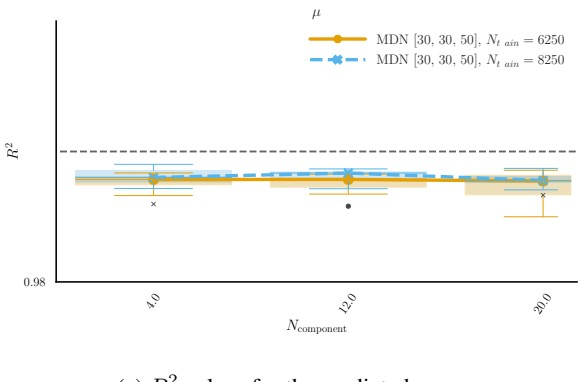

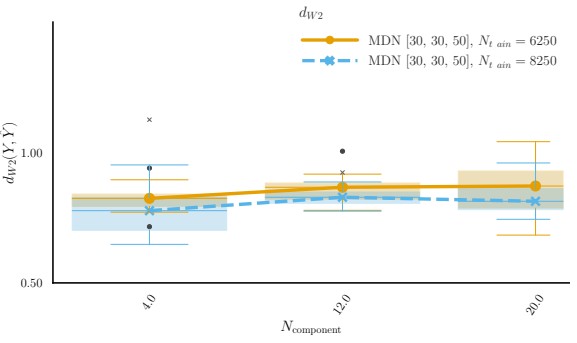

(a) $R^2$ values for the predicted mean

(b) $d_{W2}$ for the predicted conditional pdf

**Figure C2.** Sensitivity of the MDN surrogate to the number of mixture components.

mixture components. In other words, it does not necessarily benefit from a large set of mixture components. MDN models in the remainder of the study are trained with four kernels.

*Author contributions.* **Deepali Singh:** Conceptualization, Methodology, Software, Validation, Data curation, Investigation, Writing - original draft, Visualization. **Erik Haugen:** Conceptualization, Software, Data curation, Supervision, Methodology, Writing - review & editing. **Kasper Laugesen:** Conceptualization, Software, Supervision, Methodology, Writing - review & editing, Project administration. **Richard**
**P. Dwight:** Conceptualization, Supervision, Methodology, Writing - review & editing, Project administration. **Axelle Viré:** Supervision, Writing - review & editing, Project administration, Funding acquisition.

*Competing interests.* The authors declare that they have no known competing financial interests or personal relationships that could have appeared to influence the work reported in this paper.

*Acknowledgements.* The project has received funding from the European Union's Horizon 2020 research and innovation programme under
grant agreement No. 860737 (STEP4WIND project, step4wind.eu). The authors would like to thank Dr. Herman Frederik Veldkamp for his valuable insights and expert recommendations.



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
