# Peer review of "Data-driven probabilistic surrogate model for floating wind turbine lifetime damage equivalent load prediction"

_Wind Energy Science, 2025_

## Referee Comment (RC1)

*Discussion of:*

**Data-driven probabilistic surrogate model for floating wind turbine lifetime damage equivalent load prediction**

06 March 2025

The manuscript presents an application of a probabilistic surrogate modelling technique with Mixture Density Networks for prediction of floating wind turbine site-specific fatigue damage accumulation. The paper is well written and addresses relevant scientific topics such as probabilistic surrogate modelling and floating wind turbine design assessment. That being said, the scientific novelty does not become evident from the current content of the manuscript. I suggest that a significantly revised paper clearly establishes where the novelty is and focuses the narrative on the novel aspects. Please find some more elaboration in the comments below:

**General comments**

1) The manuscript discusses quite similar topics and uses similar methodologies as another recent manuscript by the same main author (https://doi.org/10.5194/wes-9-1885-2024, also cited in this paper). Please discuss what is the distinct methodological novelty of the present paper (see also my next comment).

2) As with most surrogate modelling approaches, this one is specific to the model which has been used to run the simulations. This limits the direct applicability of the trained model to just the turbine configuration in question. As a result, the primary scientific contribution of a surrogate modelling paper is normally in the methodology (or some specific findings from the results) rather than the end product. I recommend that the authors clarify what is the methodological contribution in this paper, or highlight some important findings that warrant the publication.

3) Bayesian Neural Networks (BNNs) are another approach to train a heteroscedastic model without the need of making repetitions. The authors may want to mention this and cite e.g. Hlaing et al. (https://doi.org/10.1177/14759217231186048)

4) I am missing a discussion section, which may include thoughts on the limitations of the current study.

**Specific comments**

5) References style: many references seem to introduce repetitions, such as e.g., " Zhu et al. (Zhu and Sudret, 2020) on line 65. If the authors use the \citep command to refer to a paper, they don't need to repeat the author names in the text as they come automatically from the LaTeX command.

6) Simulation time of 600s seems quite short for floating wind with low-frequency response. This may affect especially the estimation of higher-order moments of the response and may be important for this study which explicitly considers higher-order statistics.

7) Section 3.2.1: The authors suggest the R-squared between the mean and the standard deviation predictions of the distribution as a goodness-of-fit metric. This limits the representativeness of the

comparison as it doesn't allow comparing higher distribution moments. Also, the R-squared is not sensitive to bias. The other metric proposed by the authors, the Wasserstein distance, is not limited in this way. Is the R-squared then redundant? Results shown in Table 9 may hint at that, since it is only the dw2 that flags the tower bottom FA model as having worse performance than the other three channels.

8) Page 23, line 428: the authors state "Including these additional uncertainties in the feature set would likely increase the variance of the final load estimates.". I agree with this, but I think some nuance needs to be added. The uncertainties that are propagated through the fatigue model will not necessarily increase the variance of the short-term outputs, they may introduce bias in the long-term mean which will manifest as an uncertainty in the long-term (aggregated) statistics of the outputs. For example, assuming higher annual mean wind speed will lead to a bias in the mean estimate of total accumulated fatigue damage.

---

## Referee Comment (RC2)

[referee-annotated manuscript omitted]

---

## Author Comment (AC1)

**Authors' response**

Dear Editors, Reviewers,

We deeply appreciate the thorough and constructive review of the article titled "Data-driven probabilistic surrogate model for floating wind turbine lifetime damage equivalent load prediction". The detailed comments of the reviewers are extremely useful and have significantly improved the quality of the paper. Here you will find the point-by-point response to the reviewer's comments in blue. The modified/added text is quoted in the reply in italics wherever necessary. The modifications in the submitted manuscript as a result of the reviewers' comments are highlighted in blue.

In addition to the reviewers' comments, we made the following minor changes in the manuscript:

- We noticed a typo in Equation (1) with a missing summation sign in the numerator, which has been updated in the text.

- A square exponent term was missing in Equation (6), which has been corrected.

- The convergence plots Figure 9a and Figure 9b are changed from log scale in the X-axis to linear scale in Figure 10a and Figure 10b as it seems to improve the readability and clarity for the reader. The contents of the plots have not been modified.

Best regards,
Deepali Singh, Erik Haugen, Kasper Laugesen, Richard P. Dwight, Axelle Viré

**Reviewer 1**

***Q 1.1*** *The manuscript discusses quite similar topics and uses similar methodologies as another recent manuscript by the same main author (https://doi.org/10.5194/wes-9-1885-2024, also cited in this paper). Please discuss what is the distinct methodological novelty of the present paper (see also my next comment).*

**Reply**:
Indeed, the probabilistic machine learning framework used in this manuscript is the same as in the referenced article (https://doi.org/10.5194/wes-9-1885-2024). This was a deliberate decision following a thorough review of various machine learning methods for this application (Gaussian processes, conditional generative adversarial networks, chained Gaussian processes, artificial neural networks, among others). From the models we tested, Mixture Density Networks (MDNs) stood out as the most accurate, practical, and flexible. Although we do not present a comparison of different models in this paper, we would like to highlight that MDNs are particularly well-suited for probabilistic fatigue load modeling. The conditional response is often heteroscedastic and can be non-Gaussian, which the model can easily handle. In recent years, there has been a growing interest in Bayesian neural networks (as pointed out later), but we haven't evaluated it for load modeling yet. BNNs use variational inference, which can be computationally expensive, but it does allow for the separation of aleatoric and epistemic uncertainties. It would be interesting to couple BNNs with MDNs in the future to harness the advantages of both methods. We would like to highlight that the contribution of our work lies not in the ML architecture itself but in the application.

The novelty of this work is threefold:

1. **Application of probabilistic data-driven model to Floating Offshore Wind Turbines (FOWTs):**
The model is applied to FOWT DEL estimation, unlike our previous paper, where the scope was limited to fixed bottom cases. Not only are FOWT loads affected by the nonlinear dynamic behavior of the floating wind turbine, but also the underlying

simulations are much more complex, the stochastic hydrodynamic parameters have a bigger impact on the tower, the data is 'noisier', the parameter space is larger, and thereby, the number of simulations is larger. This necessitates investigations into robust surrogates. However, the literature on FOWT load modeling using probabilistic data-driven methods is currently limited to our understanding. Probabilistic data-driven models are generally validated on simpler cases. Complex cases are modeled using deterministic approaches, or with the assumption that the response is Gaussian or homoscedastic (Li and Zhang, 2019b, a). In this context, mixture models present a novel solution by addressing the limitations in the literature. They allow for uncertainty propagation, account for heteroscedasticity, and capture higher-order moments of the response distribution.

2. **Long-term probabilistic fatigue estimation with high-dimensional input**: For the lifetime fatigue estimation for FOWTs, the loads are calculated for a limited number of lumped sea states and statistically extrapolated to the lifetime loads by scaling them based on the probability of occurrence of the sea state. The number of variables is also limited, as the addition of a new variable exponentially increases the number of simulations needed for fatigue calculations. How to best choose the appropriate variables, sea states, and bin sizes to minimize errors in the extrapolated quantity is an open question (Papi and Bianchini, 2024). In the second part of the article, we use the trained surrogate model to make probabilistic estimates of fatigue loads on all available historical site data for 25 years, bypassing the need for parametric joint distribution assumptions. An extension of this study could also be to model the parameters of the distributions defining the site conditions and propagate them to the loads. Such surrogates can also be used to identify interesting sea states and bin sizes that minimize the deviation of the extrapolated fatigue compared to the reference, where all available site data is used.

3. **Insight into variance in long-term fatigue**: We demonstrate that due to the law of large numbers, the influence of random seeds on long-term tower load estimation diminishes significantly over a 25-year period, even with 500 seeds (from the surrogate). This indicates that stochastic inflow conditions contribute minimally to uncertainty in long-term fatigue predictions.

As both reviewers noted, the novelty of the work is not clearly presented in the introduction. Therefore, we have modified the introduction to separate it into "Background", "Previous work on probabilistic data-driven modeling for wind turbine load estimation", and "Research objectives". A brief overview of the surrogate modeling task is added to the Background section, instead of the end of the section. The research objective now reiterates the novelty of the work on page 5, line 425. The exact text is copied here for reference:

"*The objectives of this work are threefold:*

- *To apply and evaluate a probabilistic machine learning model (MDN) for predicting short-term fatigue loads in floating offshore wind turbines:*
  *Not only are FOWT loads affected by the nonlinear dynamic behavior of the floating wind turbine, but also the underlying simulations are much more complex, the stochastic hydrodynamic parameters have a bigger impact on the tower, the data is 'noisier', the parameter space is larger, and thereby, the number of simulations is large. This necessitates investigations into robust surrogates. Probabilistic data-driven models are often validated on simpler cases. Complex cases are modeled using deterministic approaches, or with the assumption that the response is Gaussian or homoscedastic. Using a mixture model effectively solves many of the gaps in the literature by propagating uncertainty, including heteroscedasticity, and modeling higher-order moments.*

- *To demonstrate that such a model can provide insights into long-term probabilistic fatigue estimation, especially in high-dimensional spaces where traditional binning approaches become computationally restrictive:*
  *The lifetime fatigue estimation for FOWTs involves additional complexity, including the joint distribution of metocean conditions and sensitivity to bin sizing. These issues are exacerbated by the exponential growth in simulation cost with each added dimension. A surrogate model enables direct use of historical site condition data, bypassing the need for parametric joint distribution assumptions.*

– *To investigate the impact of stochastic variability in site conditions on fatigue prediction over long time spans:*
*We select four potential floating wind sites with similar water depths and use the surrogate model to estimate probabilistic lifetime fatigue loads. We investigate the effect of accounting for the stochastic variability in the 10-minute loads relative to the variation in site conditions. It is demonstrated that due to the law of large numbers, the influence of random seeds on long-term tower load estimation diminishes significantly over a 25-year period, even with 500 seeds (from the probabilistic surrogate). This indicates that stochastic inflow conditions contribute minimally to uncertainty in long-term fatigue predictions.*

*By addressing these points, this work contributes a validated, flexible surrogate modeling framework that accounts for the complexities of floating wind systems, advancing the state of probabilistic load modeling in floating offshore wind research.*"

**Q 1.2** *As with most surrogate modelling approaches, this one is specific to the model which has been used to run the simulations. This limits the direct applicability of the trained model to just the turbine configuration in question. As a result, the primary scientific contribution of a surrogate modelling paper is normally in the methodology (or some specific findings from the results) rather than the end product. I recommend that the authors clarify what is the methodological contribution in this paper, or highlight some important findings that warrant the publication.*

**Reply**: Agree with this recommendation. Based on this comment and comment number 4, we have added a discussion section to list the key takeaways of this study, which also includes some comments on the limitations and advantages of using MDNs. Since the focus of the work is not the ML architecture itself, we have not made any changes to the methodology section.

**Q 1.3** *Bayesian Neural Networks (BNNs) are another approach to train a heteroscedastic model without the need of making repetitions. The authors may want to mention this and cite e.g. Hlaing et al. (https://doi.org/10.1177/14759217231186048)*

**Reply**: Thank you for sharing this article. It has been added to the literature review section in the introduction on page 5, line 404.

The added text is copied here for reference:
*Heteroscedastic probabilistic data-driven modeling has been further explored using Bayesian neural networks (BNNs) for estimating loads on non-instrumented wind turbines using information from fully-instrumented counterparts (Hlaing et al., 2024). Preliminary studies on fatigue load prediction using BNNs show promising results on onshore wind turbines (Omole et al., 2021). BNNs are a powerful class of neural network architectures that assign probability distributions to the network weights and biases, effectively allowing the separation of aleatoric and epistemic uncertainties.*

**Q 1.4** *I am missing a discussion section, which may include thoughts on the limitations of the current study.*

**Reply**: Agreed, a discussion section has now been included, page 29, line 833.

The exact text is quoted here for reference:

*5.2 Discussion and future work*

*This section provides a critical discussion of the study's results, along with practical considerations and limitations associated with the use of MDNs.*

*5.2.1 10-minute conditional DEL prediction*
*Given the stochastic nature of the site conditions, it is natural to model the 10-minute DEL response within a probabilistic framework. MDN is demonstrated in this study to be a reliable tool for modeling the conditional distribution of 10-minute DELs on the spar buoy floating wind turbine's tower and blades. MDN predictions are shown to remain robust across different network architectures and numbers of mixture components. The conditional means of the DELs are predicted with high*

*accuracy, achieving an $R^2 = 0.99$. Additionally, the Wasserstein distance between the predicted and reference conditional*
130 *distributions shows a strong match at the blade roots and tower top. However, at the tower bottom, the conditional standard deviation of the 10-minute fore-aft DEL is consistently over-predicted. It is corroborated by the relatively larger normalized Wasserstein distance value, indicating a bigger difference between the reference and predicted pdfs. Two main factors contribute to this: (i) the reference BHawC distributions are not converged at all simulated test locations with 44 random seeds. The tails of some distributions are not developed, resulting in short-tailed distributions that the MDN cannot easily capture;*
135 *and (ii) the tower bottom fatigue is shown in the literature to have a stronger correlation to the hydrodynamic parameters, leading to higher noise in the data. As MDN is trained to minimize the negative log-likelihood, it is rewarded for predicting higher variance when there is less confidence.*

**5.2.2 Probabilistic lifetime DEL aggregation**

140 *The uncertainty in the aggregated lifetime fatigue loads due to stochastic inputs is found to be much smaller in scale compared to the mean. This results from summing the 10-minute DELs over a million occurrences, effectively nullifying the impact of the outliers. The use of a probabilistic surrogate that correctly captures the conditional distribution is still useful, as it minimizes the aggregation of error in the final response.*

145 ### *5.2.3 Notes on mixture density networks*
*Mixture density networks, due to their flexibility in modeling the conditional response, are well-suited for the problem of probabilistic load estimation. One big advantage of the method is the ease of implementation and robustness, as demonstrated in this paper. Compared to deterministic models that often assume a Gaussian response to determine the conditional mean, mixture models can account for skewness and multimodality, and improve the mean estimates. This is especially important for*
150 *quantities like DELs, which may have non-Gaussian, heteroscedastic variations. MDNs scale well and are cost-effective to train compared to models that use Bayesian inference or variational inference (Blei et al., 2017).*

*MDNs without regularization can result in overfitting. Therefore, in this study, both L1 and L2 regularization are implemented. Secondly, MDNs rely on a stochastic optimizer that is sensitive to the initialization of the model parameters. Hence, a 10-fold*
155 *cross-validation is recommended to ensure the optimizer is not stuck on a false minimum. As seen in the tower-bottom fore-aft channel, minimizing the negative log-likelihood can result in the over-prediction of the standard deviation of the conditional response when the underlying distribution is short-tailed. MDNs here are not restricted to strictly positive values; in some cases, the tails may also extend to negative values. A potential solution is to assume a lognormal distribution for the output. This can be done by directly predicting the parameters of a lognormal distribution during training or by transforming the*
160 *output to a normal distribution before training.*

**Q 1.5** *References style: many references seem to introduce repetitions, such as e.g., " Zhu et al. (Zhu and Sudret, 2020) on line 65. If the authors use the citep command to refer to a paper, they don't need to repeat the author names in the text as they come automatically from the LaTeX command.*

**Reply**: Thanks, this has been fixed throughout the document now.

165 **Q 1.6** *Simulation time of 600s seems quite short for floating wind with low-frequency response. This may affect especially the estimation of higher-order moments of the response and may be important for this study, which explicitly considers higher-order statistics.*

**Reply**: 10-minute simulations are considered sufficient according to the IEC standards for floating wind fatigue calculations on the tower and blades, and are the standard at Siemens Gamesa for this wind turbine type. It is also in line with an extensive
170 study by Haid et al. (Hai, 2013), where the fatigue on the wind turbine tower and blades is more sensitive to whether the half-cycles are calculated than to the total duration of the simulation. Nevertheless, the optimal length of the simulation for floating wind turbines should be looked into in more detail by the research community, as the responses can be foundation-dependent. The surrogate model presented here is based on the current standards and is expected to evolve as the certification requirements

evolve. Some other components, such as the mooring lines, are more sensitive to low-frequency cycles due to their low natural frequencies and require longer simulations to capture relevant load variations. However, they are not in the scope of this article.

***Q 1.7*** *Section 3.2.1: The authors suggest the R-squared between the mean and the standard deviation predictions of the distribution as a goodness-of-fit metric. This limits the representativeness of the comparison as it doesn't allow comparing higher distribution moments. Also, the R-squared is not sensitive to bias. The other metric proposed by the authors, the Wasserstein distance, is not limited in this way. Is the R-squared then redundant? Results shown in Table 9 may hint at that, since it is only the dw2 that flags the tower bottom FA model as having worse performance than the other three channels.*

**Reply**: The $R^2$ value of the conditional mean is a simple and widely used metric that provides a reference for how good the model is in predicting the "most likely" value of the response. It is especially useful if one wants to compare it to other deterministic models that generally would only predict the conditional mean. As noted, $d_{W2}$ is introduced for a holistic comparison of the response. For the tower bottom fore-aft moment, the difference in the output of the two parameters is interesting because it indicates that even if the exact conditional distribution is not as accurately predicted as the other channels, the conditional mean is still well captured. One of the limitations in the setup is that BHawC only has a limited number of seeds, so the reference distribution is not fully converged either. It is more evident in the case of the tower bottom fore-aft moment, where the distributions in many cases are very short-tailed. The high $d_{W2}$ values are also affected by the non-converged reference distributions. Since the mean converges faster than the higher-order moments, $R^2\mu$ is not as affected.

***Q 1.8*** *Page 23, line 428: the authors state "Including these additional uncertainties in the feature set would likely increase the variance of the final load estimates.". I agree with this, but I think some nuance needs to be added. The uncertainties that are propagated through the fatigue model will not necessarily increase the variance of the short-term outputs, they may introduce bias in the long-term mean which will manifest as an uncertainty in the long-term (aggregated) statistics of the outputs. For example, assuming a higher annual mean wind speed will lead to a bias in the mean estimate of total accumulated fatigue damage.*

**Reply**: Agree with this, the text has been modified in page 27, line 789.

The exact text is quoted here for reference:

*Including these additional sources of uncertainties may introduce bias in the long-term mean, which is reflected as an uncertainty in the aggregated statistics of the outputs.*

**Reviewer 2**

***Q 2.1*** *Clarify the novelty with respect to your previous paper on Wind Energy Science. It seems that the only change is the database, while the methodology is the same.*

**Reply**: Based on this and the previous reviewer's comment on the clarity of the novelty of the work (Q1.1), more information has been added to the introduction on page 5, line 425.

***Q 2.2*** *I'm not convinced by the testing set, since it's quite specific instead of random.*

**Reply**: The test locations for some variables, like yaw misalignment and power law exponent, were chosen to be specific values based on user requirements at the time the database was being generated. We agree that, in hindsight, random sampling of all the parameters would be a fairer way to judge the performance of the surrogate. Unfortunately, creating a new test dataset is no longer possible for this study, but we do acknowledge your comment and have added a section in the text recommending

random sampling for testing in the future, page 15, line 595.

The exact text is quoted here for reference:

*For future studies, jointly sampling the test points across all variables is recommended for a fairer evaluation of the surrogate's performance throughout the domain.*

***Q 2.3*** *The literature review is well done. Have you found any paper on Bayesian neural networks? Not necessarily from wind. I'm only aware of a report from the HIPERWIND project.*

**Reply**: Thank you. We haven't looked much into BNNs, but based on the recommendation of the other reviewer, we have added a reference now to the literature review about two BNN papers by Hlaing and Omole, page 5, line 404.

***Q 2.4*** *I had troubles following the hyper-parameters tuning between the main text and the appendix. Some restructuring would be appreciated.*

**Reply**: We had decided it was necessary to move the hyperparameter tuning section to the appendix to limit the length of the article as it was getting too long.

***Q 2.5*** *Which library have you used to train the MDN?*

**Reply**: We have used tensorflow probability (https://www.tensorflow.org/probability) for modeling MDNs. The output layer is modeled using a mixture normal implementation that can be found here: https://www.tensorflow.org/probability/api_docs/python/tfp/layers/MixtureNormal.

***Q 2.6*** *Specify the names of the time-domain simulation tools.*

**Reply**: The names of some of the commonly used engineering tools are added on page 2, line 335.

The exact text is quoted here for reference:

*The calculations are typically made using time-domain multi-physics engineering tools like OpenFAST (Jonkman, 2013), HAWC2 (Larsen and Hansen, 2007), and BHawC (Couturier and Skjoldan, 2018; Skjoldan, Peter Fisker, 2011)*

***Q 2.7*** *Space between the number and the unit.*

**Reply**: A space has been introduced between the number and the unit throughout the document. Perhaps this will be standardized during the typesetting phase by the journal.

***Q 2.8*** *"The inflow turbulence is modeled using a spatially varying frozen wind field based on the Mann model." Have you tried using the hipersim turbulence generator?*

**Reply**: We only used the internal SGRE turbulence box generator for this study to keep in line with the company standards. Thanks for sharing the hipersim package, it's definitely worth looking at in the future, also within the industrial context.

***Q 2.9*** *"The structural elements are modeled using the co-rotational formulation providing geometric nonlinearity." I was expecting to read equilibrium based formulation by Philippe Coutourier*

**Reply**: The co-rotational formulation is modeled to capture geometric non-linearities, while the beam elements are modeled with a variable cross-section equilibrium formulation introduced by Coutourier. The reference to the beam model by Coutourier is added on page 9, line 483.

250   *Q 2.10 Mention the curse of dimensionality.*

**Reply**: This has been added to the text on page 10, line 500.

*Q 2.11 Addition of Winds to Loads paper by N. Dimitrov that uses Sobol indices.*

**Reply**: The article has been added to the list on page 10, line 507.

*Q 2.12 Formatting $U_{ref}$, $I_{ref}$, $n_{ref}$, $W_{dir}$, $z_{ref}$, $V_{hub}$ to remove italic subscript.*

255   **Reply**: The variable notation has been replaced to $U_{\mathrm{ref}}$, $I_{\mathrm{ref}}$, $n_{\mathrm{ref}}$, $W_{\mathrm{ref}}$, $w_{\mathrm{ref}}$, $V_{\mathrm{hub}}$ throughout the document.

*Q 2.13 Including the tower bottom side-side moment*

**Reply**: Only the fore-aft moment was included as the magnitude of fatigue in this direction is much larger than side-side. However, it could be interesting for sites with large wind-wave misalignment. The side-side moment may potentially be more difficult to model with a surrogate, as there is no aerodynamic damping of loads. Perhaps it is interesting to test the limits of 260   the surrogate and should be included in future investigations.

*Q 2.14 Non-dimensionalizing the DEL values*

**Reply**: The value of the DELs could not be shared publicly, so we decided to non-dimensionalize it using the Z-score/ standard scaler. This is now added to the text on page 11, line 534.

*Q 2.15 What is the local coordinate system?*

265   **Reply**: The local coordinate system here refers to the coordinate system of the individual component as defined in the BHawC tool and not a fixed location in the domain. But indeed, more information would be useful if we were sharing absolute load values.

*Q 2.16 Formatting of new lines after equations.*

**Reply**: The space has been removed before the beginning of the line following an equation throughout the paper.

270   *Q 2.17 Instead of fitting lower and upper bound, and then using a uniform distribution, you could have fitted these measurements with a multivariate distribution. For example, a n-dimensional normal with correlated inputs. It's quite easy with both SciPy and OpenTURNS, and you might rank which distribution performs best using the maximum likelihood.*

**Reply**: Agree, fitting a multivariate distribution and then sampling as an alternate approach preserves the marginal distribution and doesn't require an arbitrary choice of feature bounds. We have added your suggestion as an alternate approach for the 275   reader on page 14, line 584. Thanks a lot for referencing the Python libraries. We will note it for future investigations. Thanks also for introducing scrambling and the advantage of using Halton sequencing over Sobol.

The exact added text is copied here for reference:

280   *Alternatively, a multivariate distribution fitting the available data can be used to define the sampling space bounds.*

*Q 2.18 Should the wave direction be correlated to the wind direction?*

**Reply**: It is certainly correlated to the wind direction. In this case, we are not including wind direction as an input parameter, assuming the rotor is always facing the wind. So, essentially, the wave direction acts as the wind-wave misalignment angle. For floating wind loads, it is considered an important parameter, and because we wanted the model to learn to predict loads at any misalignment angle, we decided to sample it uniformly.

*Q 2.19 Please cite your previous paper with the formulas.*
*- In general, during training it helps normalizing the input and output, but it looks like here it's even more important, or you'll get a floating point overflow when computing the std. How have you normalized the database? - The softmax function can easily overflow. Are you using the trick by Nick Higham to prevent it?*

**Reply**: The previous paper is now cited in the methodology section. The data is normalized using the standard scaler/ Z-score. We have not checked if the fit changes by using the trick of subtracting $\max(x)$ from $x_i$ to prevent overflow. At least in this example, we did not encounter an overflow situation, but it is definitely something to investigate further to prevent it from happening.

*Q 2.20 How are you optimizing the hyperparameters?*

**Reply**: In this case, we only checked the performance by manually tuning selected hyperparameters. We have not looked into Optuna yet. It is a good suggestion as the number of hyperparameters grows.

*Q 2.21 Why are the number of layers and nodes in a separate table?*

**Reply**: This is to name the models with different node numbers and link them to the convergence plots in the results section.

*Q 2.22 Number of mixture components*

**Reply**: In most cases, the response does look Gaussian, but for instance in Figure 12c, the distribution is better represented with some skewness. This is only possible if we use more than one Gaussian component. Perhaps 2 or 3 components would also suffice, but having more does not necessarily increase the training cost by much while making sure there are enough parameters available to model any kind of response. In the previous paper on fixed-bottom wind turbines, we encountered multi-modal output where one component would not have captured the response correctly.

*Q 2.23 Choice of feature bounds*

**Reply**: The feature bounds are just arbitrary functions defined using trial and error, that encapsulate the observed site data. This is only an example, and perhaps using a multi-variate distribution as you suggested is a better approach to define the bounds.

[revised manuscript text omitted]

10-min. statistics - Input

Turbulence
Wind speed
Shear exponent
Yaw misalignment

Significant wave height
Spectral peak period
Wave direction

10-min. DEL - Response

Blade root edgewise and flapwise DEL

Tower bottom and tower top DEL

**Figure 1.** Schematic of the surrogate modeling objective.

[revised manuscript text omitted]

---

## Author Response (AR2)

**Authors' response**

Dear Editor, Reviewers,

We have added the citation for TensorFlow Probability in 3.1 (highlighted in blue), as requested. We agree with the second comment about Q2.22 that too many unused mixture components can indeed lead to a flatter loss function and increased epistemic uncertainty. We would like to continue investigating this aspect in the future. The implementation of methods such as the dual-step inference process (ZigZag) during training could help systematically quantify the epistemic uncertainty for a given number of mixture components. In this article, we have used regularization as a penalty for model complexity, which should prevent overfitting to a certain extent.

We thank the reviewers and the editor once again for their thorough and thoughtful review. Their feedback has not only improved the quality of this article but also provided us with valuable ideas and directions for future research.

Sincere regards,

[revised manuscript text omitted]